# BNST GABAergic neurons modulate wakefulness over sleep and anesthesia
Mengyao Li[1,8], Wen Li[2,3,8], Shanshan Liang [3], Xiang Liao [4], Miaoqing Gu[1], Huiming Li[5], Xiaowei Chen [1,6], Hongliang Liu [7] ✉, Han Qin [6] ✉ & Jingyu Xiao [7] ✉

The neural circuits underlying sleep-wakefulness and general anesthesia have not been fully investigated. The GABAergic neurons in the bed nucleus of the stria terminalis (BNST) play a critical role in stress and fear that relied on heightened arousal. Nevertheless, it remains unclear whether BNST GABAergic neurons are involved in the regulation of sleep-wakefulness and anesthesia. Here, using in vivo fiber photometry combined with electroencephalography, electromyography, and video recordings, we found that BNST GABAergic neurons exhibited arousal-state-dependent alterations, with high activities in both wakefulness and rapid-eye movement sleep, but suppressed during anesthesia. Optogenetic activation of these neurons could initiate and maintain wakefulness, and even induce arousal from anesthesia. However, chronic lesion of BNST GABAergic neurons altered spontaneous sleep-wakefulness architecture during the dark phase, but not induction and emergence from anesthesia. Furthermore, we also discovered that the BNST-ventral tegmental area pathway might participate in promoting wakefulness and reanimation from steady-state anesthesia. Collectively, our study explores new elements in neural circuit mechanisms underlying sleep-wakefulness and anesthesia, which may contribute to a more comprehensive understanding of consciousness and the development of innovative anesthetics.

The sleep-wakefulness cycle is of vital importance to mammals. It is organized in three different states, wakefulness, non-rapid eye movement (NREM) sleep, and rapid eye movement (REM) sleep[1,2]. The shift between wakefulness and sleep is a natural physiological process marked by reversible changes in consciousness. In parallel, general anesthesia (GA) also induces a state where consciousness temporarily recedes, despite being brought about by pharmacological agents[3]. GA and sleep may share overlapping subcortical mechanisms, with multiple neural clusters involved in sleep-wakefulness regulation also influencing GA[4].

The bed nucleus of the stria terminals (BNST), a major component of the extended amygdala, has been implicated in behaviors dependent on heightened arousal, such as stress, fear, reward, and anxiety[5]. It is a limbic forebrain region located relatively anterior to the thalamus, posterior to the nucleus accumbens, dorsal to the preoptic area, ventral to the lateral ventricle, laterally to the septum, and medial to the caudate/putamen and

internal capsule[6,7]. The BNST is a highly divided and heterogeneous structure containing predominantly GABAergic neurons[8,9]. Over the past few decades, there has been a surge of interest in the investigation of central GABAergic transmitter system in regulating sleep-wakefulness and GA. For instance, activation of preoptic GABAergic neurons alters sleep-wakefulness architecture, but does not regulate anesthetic induction or recovery[10]. However, activation of GABAergic neurons in the lateral septum, lateral hypothalamus, and the nucleus accumbens not only promotes wakefulness but also facilitates emergence from anesthesia[11–13]. These investigations indicate that inhibitory GABAergic neurons in different nuclei play distinct roles in the regulation of anesthesia, sleep, and wakefulness. Despite these findings, the precise role of BNST GABAergic neurons in the regulation of sleep-wakefulness and GA remains to be elucidated.

In clinical research, patients with refractory obsessive-compulsive disorder often experience insomnia following deep brain stimulation

[1]Advanced Institute for Brain and Intelligence, School of Medicine, Guangxi University, Nanning 530004, China. [2]Department of Neurology, Daping Hospital, Third Military Medical University, Chongqing 400042, China. [3]Brain Research Center and State Key Laboratory of Trauma, Burns, and Combined Injury, Third Military Medical University, Chongqing 400038, China. [4]Center for Neurointelligence, School of Medicine, Chongqing University, Chongqing 400044, China. [5]Department of Anesthesiology and Perioperative Medicine, Xijing Hospital, Fourth Military Medical University, Xi'an 710032 Shaanxi, China. [6]Chongqing Institute for Brain and Intelligence, Guangyang Bay Laboratory, Chongqing 400064, China. [7]Department of Anesthesiology, Chongqing University Cancer Hospital, Chongqing 400030, China. [8]These authors contributed equally: Mengyao Li, Wen Li. ✉e-mail: liuhl75@163.com; qinhan66@foxmail.com; jyxiao1989@cqu.edu.cn

targeting the BNST[14]. And basic studies have reported that stimulation of BNST GABAergic population in *Gad67-Cre* mice is able to generate an immediate transition from NREM sleep to wakefulness[15]. A recent study has demonstrated that the prepronociceptin-expressing BNST neurons, which is a subpopulation of GABAergic neurons, are involved in arousal circuitry[16]. Moreover, the level of c-Fos expression has been changed in the BNST after sleep deprivation or anesthesia[17,18]. Based on these combined findings, we hypothesize that BNST GABAergic neurons may be involved in the regulation of sleep-wakefulness and GA. However, the activity pattern and specific role of BNST GABAergic neurons, along with their downstream mechanisms in sleep-wakefulness transition and anesthesia regulation remain unclear.

To fully understand the role of BNST GABAergic neurons in sleep-wakefulness and GA, we performed a series of experiments in *Vgat-Cre* mice. It revealed that BNST GABAergic neurons displayed arousal-state-dependent alterations in population $Ca^{2+}$ signals, with high activity in both wakefulness and REM sleep. In addition, the $Ca^{2+}$ signals of BNST GABAergic neurons were significantly suppressed during isoflurane anesthesia, and recovered as isoflurane washed out. Then, we identified that optogenetic stimulation of these neurons rapidly induced a transition from NREM or REM sleep into wakefulness. A more prolonged stimulation produced sustained wakefulness, even during a period of elevated sleep drive. And photostimulation of BNST GABAergic neurons robustly induced arousal from steady-state or deep anesthesia. Ablation of BNST GABAergic neurons alters spontaneous sleep-wakefulness architecture during the dark phase, but not induction and emergence from anesthesia. What's more, we demonstrated that photostimulation of GABAergic axon terminals of BNST in the ventral tegmental area (VTA) not only promoted wakefulness from natural sleep but also reanimation from steady-state anesthesia. However, this wake-promoting effect was not observed in the deep anesthesia state. Taken together, our current study provides evidence for BNST GABAergic neurons in modulating wakefulness over sleep and anesthesia, possibly in part through VTA downstream pathway.

## Results

### Population activities of BNST GABAergic neurons across sleep-wakefulness cycle

To investigate the real-time activities of BNST GABAergic neurons across natural sleep-wakefulness cycle in vivo, we combined cell-type specific fiber photometry with electroencephalography (EEG), electromyography (EMG), and video recordings in *Vgat-Cre* mice (Fig. 1a, the verification of transgenic mouse in Supplementary Fig. 1a, b). Briefly, an optical fiber was implanted just above the BNST three weeks after injection of AAV encoding Cre-inducible $Ca^{2+}$ indicator GCaMP6f into the BNST of *Vgat-Cre* mice (Fig. 1b). EEG-EMG electrodes were simultaneously inserted into the cortical surface and neck muscles. The expression of $Ca^{2+}$ indicator GCaMP6f and fiber tip location were confirmed by *post hoc* histology (Fig. 1c). Mice were allowed to recover for at least 7 days, prior to $Ca^{2+}$ signals recording. The sleep-wakefulness states were identified by synchronous EEG-EMG signals. Furthermore, no fluctuations were detected in EGFP-labeled GABAergic neurons of the BNST, suggesting that $Ca^{2+}$ dynamics were states-dependent and not influenced by movement artifacts (Supplementary Fig. 1c–f).

To quantify $Ca^{2+}$ activity across different states, the $Ca^{2+}$ signals were averaged over the duration of wake, NREM, or REM sleep episodes. (Fig. 1d). The $Ca^{2+}$ signals during wakefulness were significantly higher than NREM sleep (Fig. 1e, wake vs. NREM: $5.63 \pm 1.16\%$ vs. $3.17 \pm 0.55\%$, $P = 0.0373$), and significantly greater during REM sleep (Fig. 1e, wake vs. REM: $5.63 \pm 1.16\%$ vs. $8.88 \pm 1.98\%$, $P = 0.0382$). In addition, dynamic profiles of BNST GABAergic neurons during different sleep-wakefulness state transitions were further analyzed. The activities of BNST GABAergic neurons increased from NREM sleep to wakefulness (Fig. 1f, NREM vs. wake: $2.14 \pm 0.44\%$ vs. $4.75 \pm 1.09\%$, $P = 0.0134$) and decreased from wakefulness to NREM sleep (Fig. 1f, wake vs. NREM: $3.74 \pm 0.79\%$ vs. $2.27 \pm 0.43\%$, $P = 0.012$). Additionally, an increased $Ca^{2+}$ activity from

NREM sleep to REM sleep (Fig. 1f, NREM vs. REM: $2.79 \pm 0.57\%$ vs. $8.99 \pm 2.38\%$, $P = 0.0196$), and a decreased $Ca^{2+}$ activity from REM sleep to wakefulness (Fig. 1f, REM vs. wake: $8.08 \pm 1.98\%$ vs. $2.98 \pm 0.53\%$, $P = 0.0191$) were observed. Overall, these findings indicate that BNST GABAergic neurons are highly active during both wakefulness and REM sleep and display arousal-state-dependent alterations in population activities. We observed that BNST GABAergic neurons are highly active during both wakefulness and REM sleep.

### Anesthetics suppress the population activities of BNST GABAergic neurons

Next, we investigated the impact of GA on the population activities of BNST GABAergic neurons. The changes in fluorescence intensity of BNST GABAergic neurons expressing GCaMP6f were measured by fiber photometry combined with EEG-EMG electrodes during exposure to 1.4% isoflurane or pure oxygen (Fig. 2a). Example traces demonstrated that the fluorescence intensity dropped upon administration of isoflurane, and gradually returned after isoflurane was turned off, and recovered to similar levels of pre-isoflurane as isoflurane washed out (Fig. 2b). Compared with $Ca^{2+}$ fluorescence intensity levels before isoflurane anesthesia, population activities of BNST GABAergic neurons were strongly inhibited during exposure to 1.4% isoflurane (Fig. 2c, Pre-Iso vs. During-Iso: $0.13 \pm 0.29$ vs. $-6.16 \pm 0.57$, $P = 0.0001$). After isoflurane washing out, $Ca^{2+}$ signals intensity of these neurons rapidly returned (Fig. 2c, During-Iso vs. Post-Iso: $-6.16 \pm 0.57$ vs. $-0.68 \pm 0.64$, $P = 0.00004$). To further validate the effects of isoflurane on BNST GABAergic neurons at the single-cell level, we applied whole-cell patch clamp recordings. Indeed, bath application of isoflurane onto slices significantly reduced the firing rate of BNST GABAergic neurons, which returned to pretreatment levels after washout of isoflurane (Supplementary Fig. 2). Collectively, our study demonstrates that the activities of BNST GABAergic neurons are strongly suppressed by isoflurane.

### Optogenetic activation of BNST GABAergic neurons promotes wakefulness

We then explored if activation of BNST GABAergic neurons was sufficient to initiate wakefulness from natural sleep. To begin, we stereotaxically injected AAV-DIO-ChR2-mCherry bilaterally into the BNST of *Vgat-Cre* mice. After three weeks of virus expression, EEG-EMG electrodes and optical fibers were concurrently implanted (Fig. 3a). Representative imaging demonstrated that ChR2-mCherry was expressed in BNST GABAergic neurons (Fig. 3b). In vitro electrophysiology recording verified that the ChR2-expressing BNST GABAergic neurons were activated by light stimulations (Fig. 3c). A typical example indicated that photostimulation of BNST GABAergic neurons triggered a rapid transition from stable NREM sleep to wakefulness (Fig. 3d, upper panel). We subsequently analyzed the probabilities of sleep-wakefulness before, during, and after photostimulation of BNST GABAergic neurons (Fig. 3e). The probability of wakefulness increased (Fig. 3f, mCherry vs. ChR2: $10.04 \pm 1.74\%$ vs. $98.38 \pm 0.78\%$, $P = 0.0002$) rapidly during activation, accompanied by a corresponding suppression of NREM sleep (Fig. 3f, mCherry vs. ChR2: $90.92 \pm 1.73\%$ vs. $1.62 \pm 0.78\%$, $P = 0.0002$). The latency from NREM sleep to wakefulness was significantly shortened (Fig. 3g, mCherry vs. ChR2: $56.78 \pm 4.24$ s vs. $1.21 \pm 0.21$ s, $P = 0.0002$) compared to the mCherry control. Moreover, light-simulation of these neurons during stable REM sleep reliably induced transitions to wakefulness (Fig. 3d, lower panel). Within 20 s of light-simulation, the probability of wakefulness increased (Fig. 3h, i, mCherry vs. ChR2: $15.31 \pm 5.33\%$ vs. $84.19 \pm 3.85\%$, $P = 0.0002$) alongside a decrease in REM sleep probability (Fig. 3h, i, mCherry vs. ChR2: $84.12 \pm 5.61\%$ vs. $15.81 \pm 3.85\%$, $P = 0.0002$). Similarly, excitation of BNST GABAergic neurons strongly reduced latency from REM sleep to wakefulness (Fig. 3j, mCherry vs. ChR2: $55.30 \pm 10.35$ s vs. $3.87 \pm 0.79$ s, $P = 0.0002$).

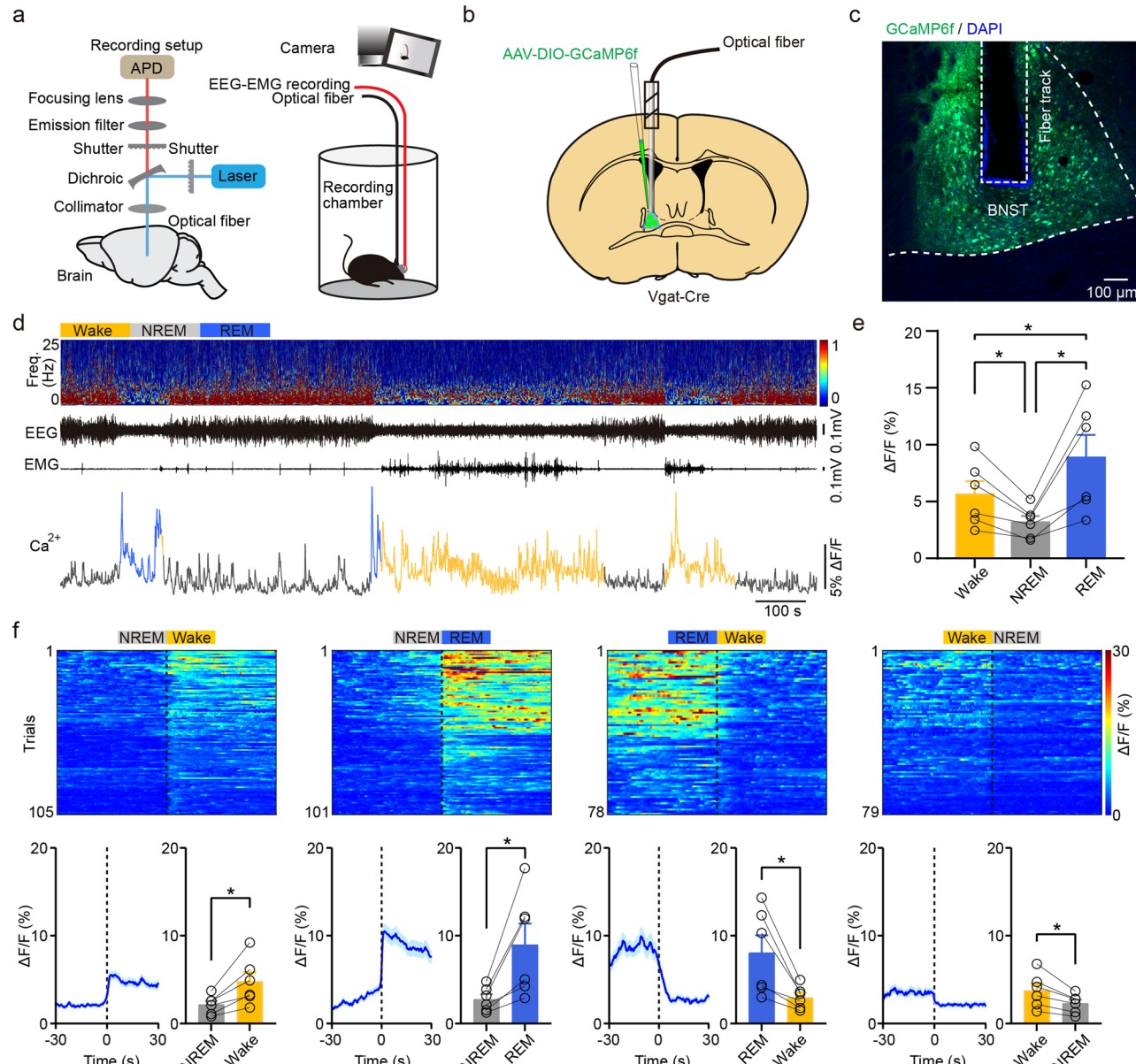

**Fig. 1 | BNST GABAergic neurons are highly active during wakefulness and REM sleep. a** Left: schematic of fiber photometry recording devices. Right: experimental design of synchronized $Ca^{2+}$ signals, EEG-EMG signals, and behavior video recordings. APD: avalanche photo diode. **b** Diagram of injection of AAV-DIO-GCaMP6f into the BNST of *Vgat-Cre* mice. **c** Confocal image illustrating virus expression (green) and fiber tip location in the BNST with DAPI (blue) as the counterstain. **d** Representative EEG power spectrum, $Ca^{2+}$ signals, and EMG traces during natural sleep–wakefulness states. Yellow indicates wakefulness, gray indicates NREM sleep, and blue indicates REM sleep. Freq.: frequency. **e** Quantification of $Ca^{2+}$ activity during wakefulness, NREM sleep, and REM sleep. $n = 6$ mice. **f** Top: heatmaps depicting individual recording traces aligned to sleep–wakefulness state transitions. NREM sleep to wake (105 transitions from 6 mice), NREM sleep to REM sleep (101 transitions from 6 mice), REM sleep to wake (78 transitions from 6 mice), and wake to NREM sleep (79 transitions from 6 mice). Bottom: average of all recording traces and statistical analysis from 30 s before and 30 s after state transitions. All transitions expressed as mean (blue) ±SEM (shaded). $n = 6$ mice. Data are presented as mean ± SEM. Statistical comparisons were determined using one-way RMs ANOVA with Bonferroni *post hoc* test or paired Student's *t* test. *$P < 0.05$.

We next examined whether BNST GABAergic neurons contributed to maintenance of wakefulness. Prolonged optical stimulation (10 ms of 473 nm pulses at 20 Hz, 20 s on / 40 s off for 60 cycles) was delivered into BNST GABAergic neurons between 10:00 am and 11:00 am (Supplementary Fig. 3a, b). Sustained activation of these neurons significantly increased the amount of wakefulness (Supplementary Fig. 3c, $P_{between\ group} = 0.0008$) and decreased the amount of NREM sleep in ChR2-mCherry mice (Supplementary Fig. 3d, $P_{between\ group} = 0.0085$) during optical stimulation, but not significantly affect REM sleep (Supplementary Fig. 3e). Overall, these results indicate that optogenetic activation of BNST GABAergic neurons is sufficient to initiate and maintain wakefulness.

## Photostimulation of BNST GABAergic neurons induces behavioral arousal during steady-state anesthesia and EEG arousal during deep anesthesia

It was reported that photostimulation of wake-promoting brain areas induced rapid emergence from GA[11,12,19]. Therefore, we tested if activation of BNST GABAergic neurons may also induce reanimation from GA. We found that optogenetic activation of BNST GABAergic neurons expressing ChR2-mCherry caused immediate behavioral arousal defined by desynchronized low amplitude EEG, high EMG activity (Fig. 4a) and body movement (Supplementary Fig. 4a and Supplementary Table 1) during steady-state anesthesia. The power density of EEG signals was analyzed across different frequency bands from 0 to 25 Hz of each 0.2 Hz bin (Fig. 4b).

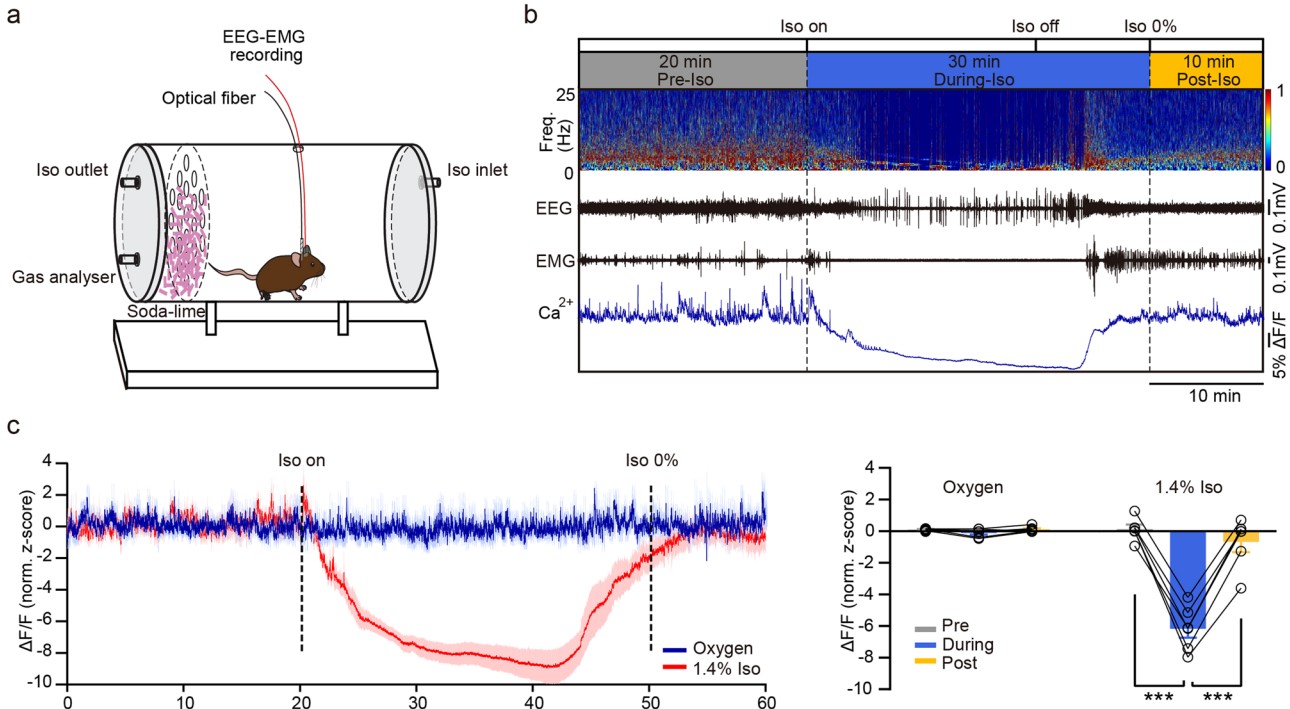

**Fig. 2 | Suppression of population activities of BNST GABAergic neurons by isoflurane. a** Experimental setup for fiber photometry recording combined with EEG-EMG recordings during isoflurane-induced general anesthesia. Iso: isoflurane. **b** Timeline of the experiment and example showing EEG power spectrum, raw EEG-EMG signals and $Ca^{2+}$ signals. Freq.: frequency. **c** Left: Time courses of average $Ca^{2+}$ activity expressed as mean (red for 1.4% isoflurane and blue for pure oxygen) ±SEM (shaded). Right: Quantification of $Ca^{2+}$ activity change pre, during, and post iso-flurane inhalation. $n = 6$ mice in each group. Data are presented as mean ± SEM. Statistical comparisons were determined using one-way RMs ANOVA with Bon-ferroni *post hoc* test. ***$P < 0.001$.

And quantitative spectral analysis of EEG signals indicated that photo-stimulation of BNST GABAergic neurons significantly triggered a decrease in the relative power of delta bead (Fig. 4c, Before vs. Stim: 50.01 ± 1.32% vs. 32.53 ± 1.78%, $P < 0.001$), but an increase in alpha band (Fig. 4c, Before vs. Stim: 10.58 ± 0.47% vs. 15.82 ± 0.76%, $P = 0.0018$) and beta band (Fig. 4c, Before vs. Stim: 6.53 ± 0.31% vs. 16.89 ± 1.38%, $P < 0.001$). However, there were no significant changes in mCherry mice with 20 Hz photostimulation (Fig. 4d–f).

We then explored the effect of photostimulation of BNST GABAergic neurons on restoring states of consciousness during burst-suppression oscillations induced by deep anesthesia. Optogenetic activation of BNST GABAergic neurons expressing ChR2-mCherry rapidly induced an EEG activity shift from burst-suppression oscillations toward an EEG arousal, without altering EMG activity (Fig. 4g), which was similar to the results of activation of GABAergic neurons in the nucleus accumbens and dor-somedial hypothalamus[12,20]. The ChR2 mice exhibited a robust increase in burst duration during photostimulation compared to the 60 s before sti-mulation (Fig. 4h, Before vs. Stim: 26.44 ± 3.03 s vs. 43.14 ± 4.23 s, $P = 0.0005$). This was accompanied by a decreased burst suppression ratio (BSR) during photostimulation compared to 60 s before stimulation (Fig. 4i, Before vs. Stim: 51.60 ± 5.17% vs. 22.67 ± 6.83%, $P = 0.0001$). In contrast, there were no significant differences in mCherry mice (Fig. 4j–l). These results illustrate that activation of BNST GABAergic neurons is capable of driving behavioral arousal during steady-state anesthesia and EEG arousal during deep anesthesia.

**Ablation of BNST GABAergic neurons alters spontaneous sleep-wakefulness architecture during the dark phase, but not induction and emergence from anesthesia**

To test the necessity of BNST GABAergic neurons in promoting natural wakefulness, we genetically ablated BNST GABAergic neurons by bilaterally injecting AAV-DIO-taCaspase3 (AAV-DIO-mCherry for control) into the BNST of *Vgat-Cre* mice (Fig. 5a). The targeted brain lesions were confirmed by neuron-specific nuclear-binding protein (NeuN) immunofluorescent staining (Fig. 5b). Hourly percentage of time spent in each state over a 24-h-recording was shown in Fig. 5c. During the dark phase, lesion of BNST GABAergic neurons decreased the amount of wakefulness (Fig. 5d, mCherry vs. Caspase3: 68.80 ± 1.87% vs. 45.08 ± 1.96%, $P = 0.0002$), and increased the amount of NREM sleep (Fig. 5d, mCherry vs. Caspase3: 29.04 ± 1.70% vs. 50.31 ± 1.76%, $P = 0.0002$) and REM sleep (Fig. 5d, mCherry vs. Caspase3: 2.16 ± 0.35% vs. 4.63 ± 0.48%, $P = 0.0034$). During the light phase, there was no significant difference in the amount of wake-fulness between two groups (Fig. 5e). Similarly, lesion of BNST GABAergic neurons decreased the amount of wakefulness (Fig. 5f, mCherry vs. Cas-pase3: 52.68 ± 1.67% vs. 41.00 ± 2.03%, $P = 0.0026$), and increased the amount of NREM sleep (Fig. 5f, mCherry vs. Caspase3: 42.14 ± 1.74% vs. 53.49 ± 1.80%, $P = 0.0006$), but did not alter the amount REM sleep over a 24 h period. We next investigated whether ablation of BNST GABAergic neurons altered behavioral consequences of isoflurane-induced GA. The induction and emergence time were used to evaluate the hypnotic properties of isoflurane in mice. To our surprise, there was no significant difference between the mCherry and lesion group in the induction and emergence time with isoflurane exposure (Fig. 5g, h).

**Photostimulation of GABAergic axon terminals of BNST in the VTA promotes wakefulness from natural sleep and reanimation from steady-state anesthesia**

The VTA is a key node in the regulation of sleep-wakefulness, which receives dense inputs from BNST GABAergic neurons[6,21]. Therefore, we investigated the role of activation of BNST GABAergic axon terminals in the VTA during both sleep and anesthesia. Three weeks after AAV-DIO-ChR2-mCherry was transduced unilaterally in the BNST, an optical fiber was ipsilaterally inserted directly above the downstream VTA alongside with EEG-EMG electrodes which were inserted into the cortical surface

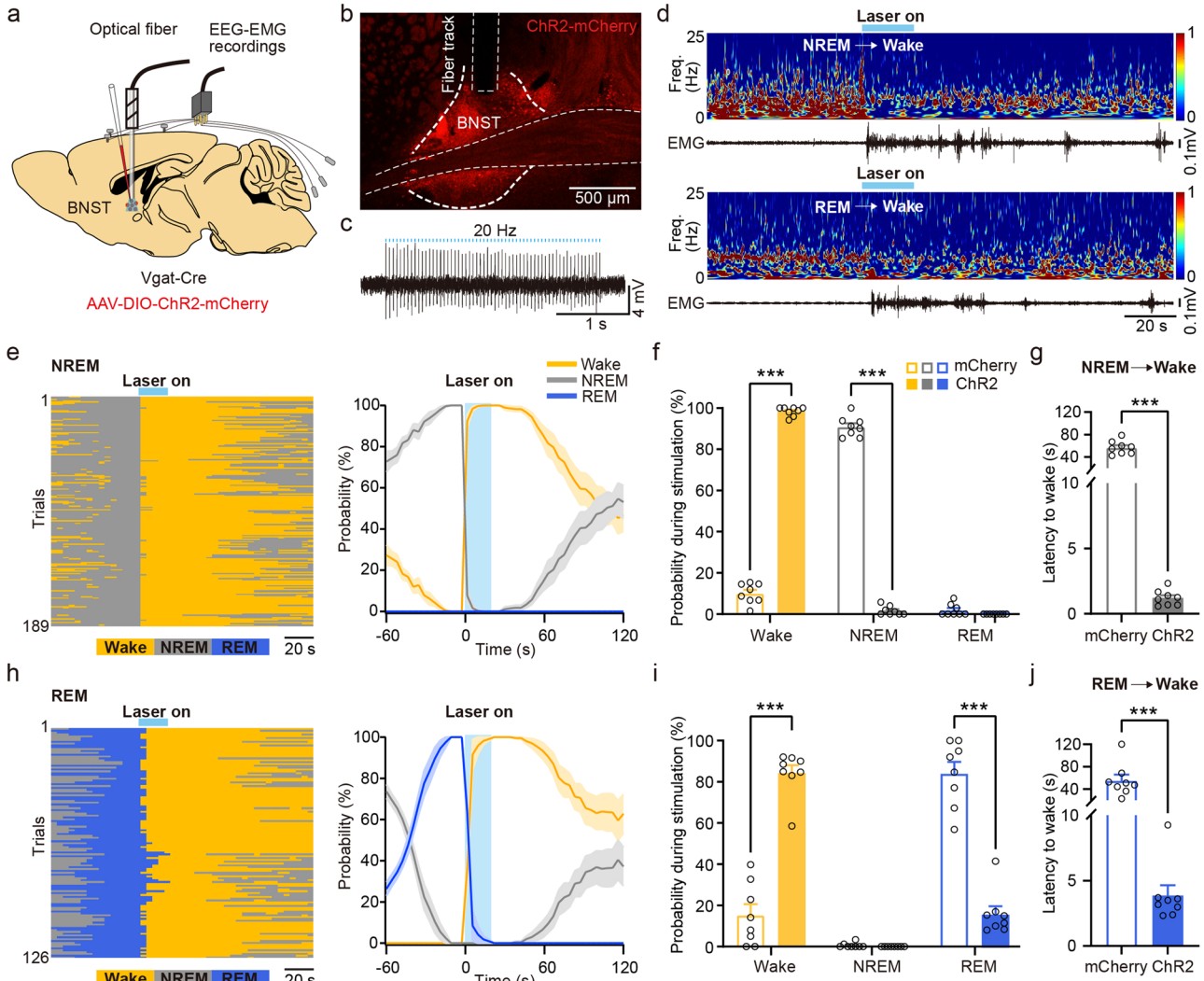

**Fig. 3 | Photostimulation of BNST GABAergic neurons triggers wakefulness from NREM and REM sleep. a** Schematic of bilateral optogenetic manipulation of BNST GABAergic neurons and simultaneous EEG-EMG recordings in *Vgat-Cre* mice. **b** An example photograph illustrating BNST GABAergic neurons expressing ChR2-mCherry and the track of the optical fiber implanted just above the BNST. **c** Representative trace of a cell-attached recording in ChR2-expressing neuron in BNST evoked by 473 nm light stimulation (3 s stimulation train at 20 Hz; 10 ms light pulses). Blue bar indicates light pulse train. *n* = 3 mice. **d** Typical examples of photostimulation of BNST GABAergic neurons during stable NREM sleep (upper) and REM sleep (lower). Blue bands, 473 nm laser stimulation (20 Hz, 10 ms, 20 s). Freq.: frequency. **e** Left: brain states of each trial in the ChR2 group undergoing photostimulation during stable NREM sleep. Yellow indicates wakefulness, gray indicates NREM sleep, and blue indicates REM sleep. *n* = 189 trials from 8 mice. Right: probabilities of each brain state before, during, and after photostimulation of BNST GABAergic neurons during stable NREM sleep. **f** Quantification of probabilities of each brain state within 20 s of light-simulation. *n* = 8 mice in each group. **g** The latency from NREM sleep to wakefulness. *n* = 8 mice in each group. **h** Left: brain states of each trial in the ChR2 group during photostimulation at stable REM sleep. *n* = 126 trials from 8 mice. Right: probabilities of each brain state before, during, and after photostimulation of BNST GABAergic neurons during stable REM sleep. **i** Quantification of probabilities of each brain state within 20 s of light-simulation. *n* = 8 mice for each group. **j** The Latency from REM sleep to wakefulness in each group. *n* = 8 mice in each group. Data are presented as mean ± SEM. Statistical comparisons were determined using Wilcoxon rank-sum test. ***P < 0.001.

and neck muscles. Histological results illustrated the expression of ChR2-mCherry in BNST GABAergic neurons and the location of the optical fiber in the VTA (Fig. 6a). The optogenetic intensity was set at 5 mW, a level which excluded the possible confounding consequences of antidromic activation in BNST cell bodies (Supplementary Fig. 5). We found that photostimulation of projections in the VTA from the ChR2-positive BNST GABAergic neurons induced immediate wakefulness during stable NREM sleep or REM sleep (Fig. 6b). The latency from NREM sleep to wakefulness (Fig. 6c, mCherry vs. ChR2: 63.53 ± 8.26 s vs. 1.25 ± 0.42 s %, *P* = 0.0079) and from REM sleep to wakefulness (Fig. 6c, mCherry vs. ChR2: 50.96 ± 6.74 s vs. 6.83 ± 1.26 s, *P* = 0.0079) was significantly shortened.

During steady-state anesthesia, optogenetic activation of these axon terminals in the VTA promptly elicited behavioral arousal in the ChR2

group, as assessed by EEG-EMG activity and arousal scores (Fig. 6d, Supplementary Fig. 4b and Supplementary Table 2). The power density of EEG signals was shown across different frequency bands from 0 to 25 Hz (Fig. 6e). A significant decrease in the relative power of delta band (Fig. 6f, Before vs. Stim: 58.46 ± 2.30% vs. 52.83 ± 1.78%, *P* < 0.0005), but an increases in alpha band (Fig. 6f, Before vs. Stim: 7.9 ± 0.59% vs. 11.55 ± 0.82%, *P* < 0.0366) was observed in the ChR2 group. However, this activation resulted in no impact in mCherry group (Fig. 6g–i). What's more, there were no indications of EEG arousal from burst-suppression oscillations and no significant EEG spectral changes induced by deep anesthesia in both groups (Fig. 6j–o). Ultimately, we identify the VTA as a potential BNST GABAergic downstream pathway in the modulation of wakefulness and emergence from steady-state anesthesia, but not emergence from deep anesthesia.

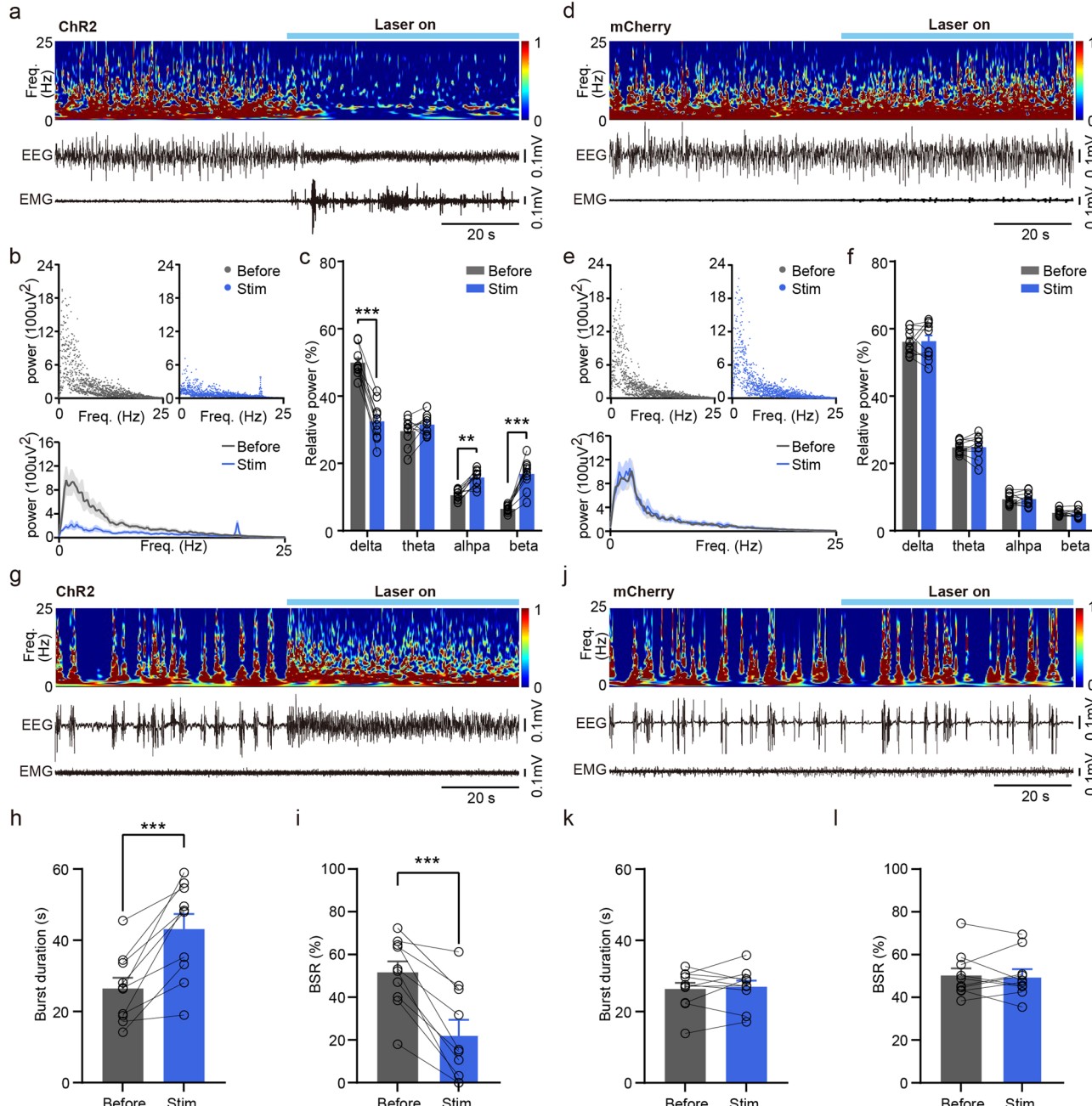

**Fig. 4 | Optogenetic activation of BNST GABAergic neurons induces behavioral arousal during steady-state anesthesia and EEG arousal during deep anesthesia.** **a** An example trace illustrating representative EEG power spectrum and raw EEG-EMG data surrounding optogenetic stimulation during isoflurane-induced steady-state anesthesia in the ChR2 group. **b** Upper: EEG power of each 0.2 Hz bin from 0 to 25 Hz 60 s before and during light-stimulation in the ChR2 group. Lower: average EEG power of different frequency bands expressed as mean (gray for 60 s before light-stimulation and blue for during light-stimulation) ±SEM (shaded). **c** Quantitative of relative power of EEG signals across four bands (delta: 0.5–4 Hz, theta: 4–10 Hz, alpha: 10–15 Hz, beta: 15–25 Hz) before and during optogenetic stimulation in the ChR2 group. $n$ = 10 trials from 5 mice. **d** An example trace illustrating representative EEG power spectrum and raw EEG-EMG data surrounding optogenetic stimulation during isoflurane-induced steady-state anesthesia in the mCherry group. **e** Upper: EEG power of each 0.2 Hz bin from 0 to 25 Hz 60 s before and during light-stimulation in the mCherry group. Lower: average EEG power of different frequency bands expressed as mean (gray for 60 s before light-

stimulation and blue for during light-stimulation) ±SEM (shaded). **f** Quantitative of relative power of EEG signals across four bands before and during optogenetic stimulation in the mCherry group. $n$ = 10 trials from 5 mice. **g** An example trace illustrating representative EEG power spectrum and raw EEG-EMG data surrounding optogenetic stimulation during burst-suppression oscillations induced by deep anesthesia in the ChR2 group. **h** Quantification of burst duration before and during photostimulation in the ChR2 group. $n$ = 10 trials from 5 mice. **i** BSR before and during photostimulation in the ChR2 group. $n$ = 10 trials from 5 mice. **j** An example trace illustrating representative EEG power spectrum and raw EEG-EMG data surrounding optogenetic stimulation during burst-suppression oscillations induced by deep anesthesia in the mCherry group. **k** Quantification of burst duration before and during photostimulation in the mCherry group. $n$ = 10 trials from 5 mice. **l** BSR before and during photostimulation in the mCherry group. $n$ = 10 trials from 5 mice. Data are presented as mean ± SEM. Statistical comparisons were determined using two-way RMs ANOVA with Bonferroni *post hoc* test or paired Student's *t* test. ***$P$ < 0.001.

**Fig. 5 | Ablation of BNST GABAergic neurons alters spontaneous sleep-wakefulness architecture during the dark phase, but not induction and emergence from anesthesia. a** Diagram showing the injection of AAV-DIO-taCaspase3 or AAV-DIO-mCherry into the BNST of *Vgat-Cre* mice. **b** Representative images of NeuN immunofluorescence staining in the BNST. **c** Time-course of wakefulness, NREM and REM sleep during 24 h sleep-wakefulness cycle. **d-f** Quantitative analysis of time spent in each state during the dark phase, light phase and 24 h sleep-wakefulness cycle. **g**, **h** Induction and emergence time exposed to 1.4% isoflurane. *n* = 8 mice in each group. Data are presented as mean ± SEM. Statistical comparisons were determined using two-way RMs ANOVA with Bonferroni *post hoc* test or Wilcoxon rank-sum test. **\*\****P* < 0.01, \*\*\**P* < 0.001.

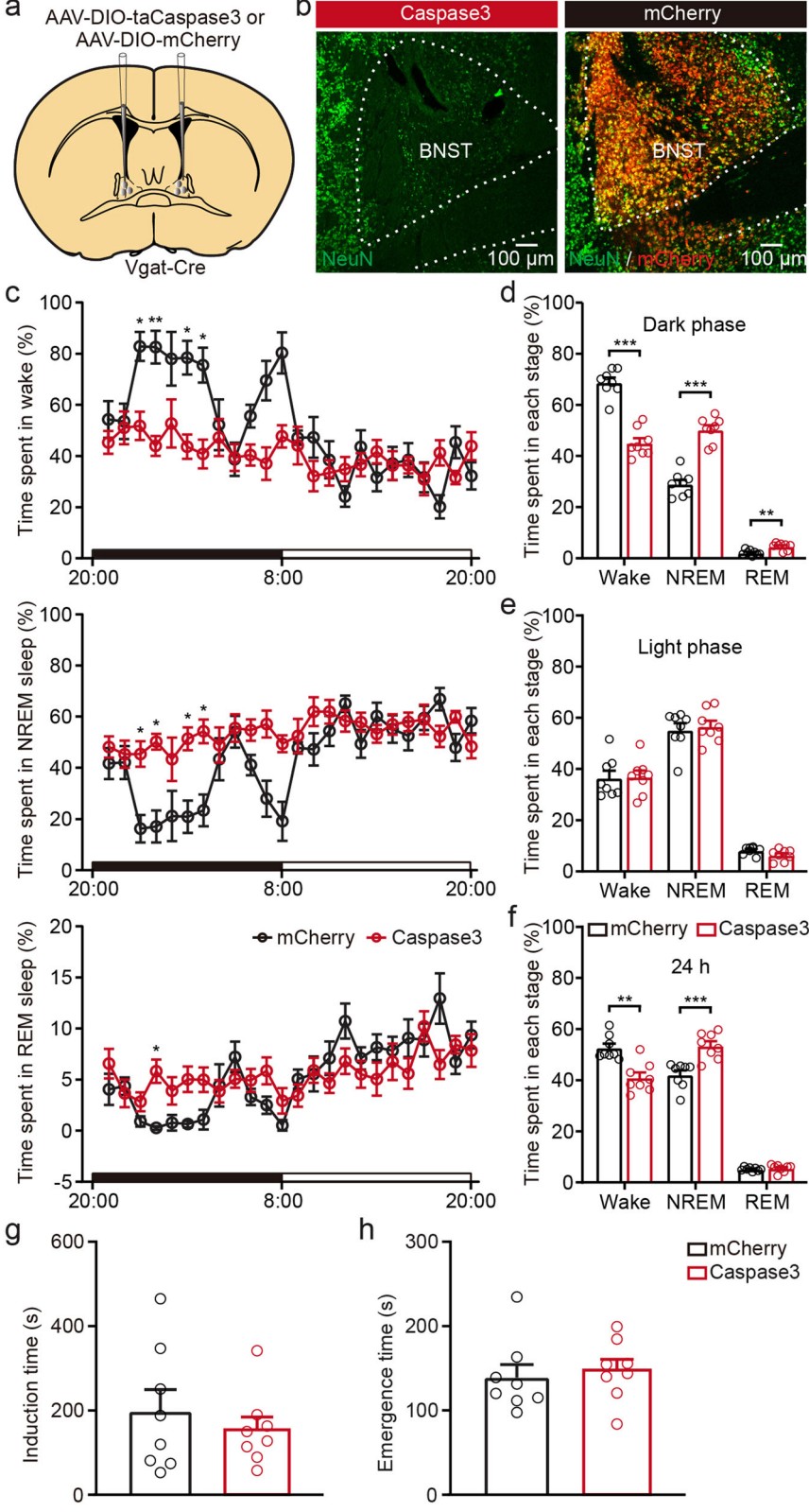

## Discussion

The neural circuits governing sleep-wakefulness and GA remain incompletely understood. The BNST contains predominantly GABAergic neurons and plays a key role in the regulation of anxiety and fear-related behavior that relied on heightened arousal in both humans and rodents[8,22,23]. Evidence suggests that the BNST GABAergic neurons may be involved in sleep-wakefulness transition and anesthesia regulation, but further clarification is needed to understand the precise role of these neuronal clusters. In the present study, we transduced the BNST of *Vgat-Cre* mice with Cre-dependent GCaMP6f virus to characterize the population activity of BNST GABAergic neurons. These neurons were highly active during wakefulness and REM sleep, with robust alterations

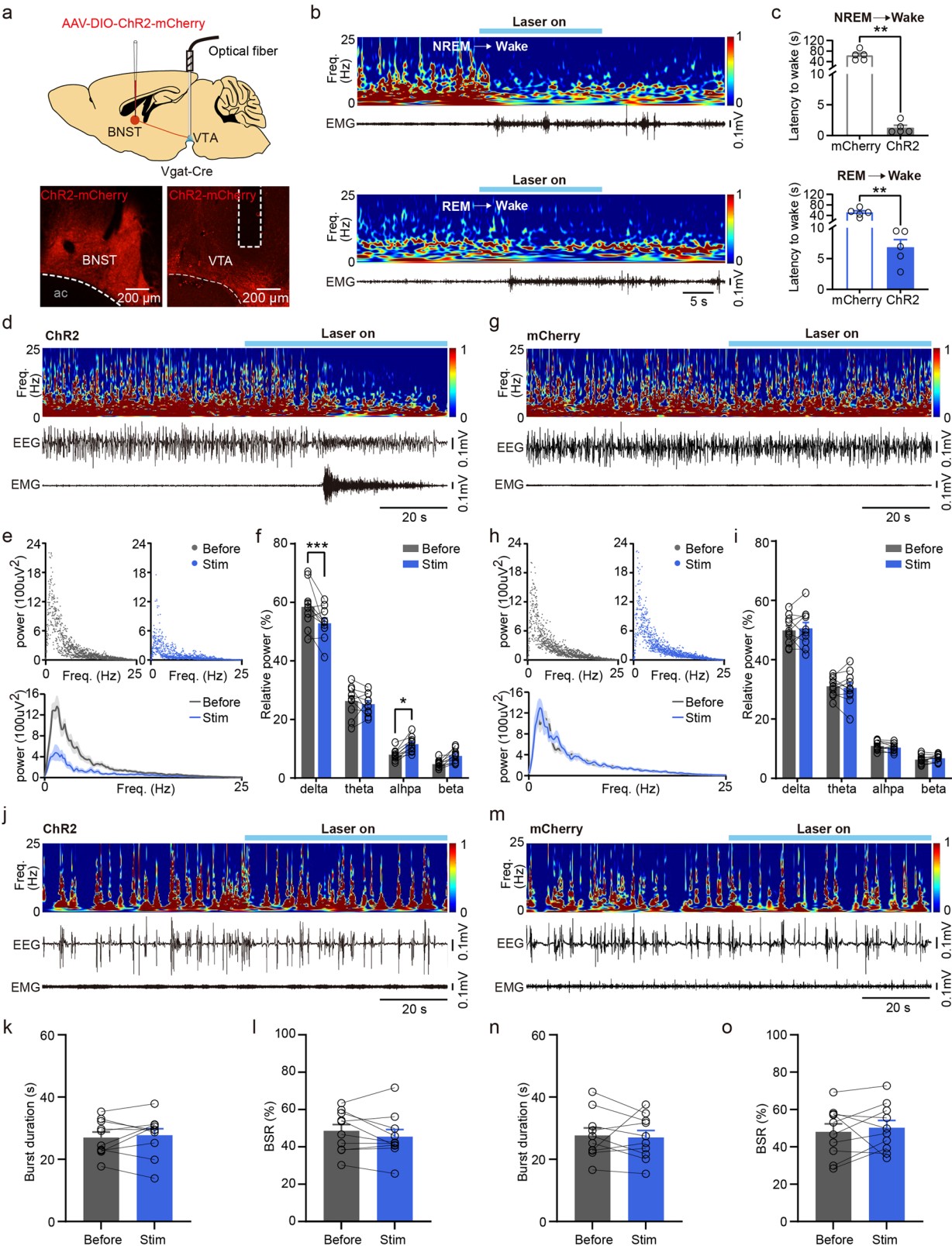

across sleep-wakefulness states. Additionally, fiber photometry and electrophysiological recordings demonstrated that BNST GABAergic neurons were inhibited during application of isoflurane and returned after isoflurane washing out. Chronic lesion of BNST GABAergic neurons altered spontaneous sleep-wakefulness architecture during the dark phase, but not induction and emergence from anesthesia. Optogenetic stimulation of BNST GABAergic neurons and the downstream VTA pathway rapidly induced a transition from NREM or REM sleep into wakefulness, as well as reanimation from steady-state GA. Overall, our study indicated that BNST GABAergic neurons could modulate wakefulness over sleep and anesthesia, possibly in part through the VTA descending pathway.

**Fig. 6 | Activation of BNST GABAergic axon terminals in the VTA over sleep and anesthesia. a** Top: schematic of optogenetic manipulation of BNST GABAergic axon terminals in unilateral VTA and EEG-EMG recordings in *Vgat-Cre* mice. Bottom: Example images illustrating BNST GABAergic neurons expressing ChR2-mCherry and the location of the optical fiber in the VTA. **b** Representative examples showing photostimulation of BNST GABAergic axon terminals in the VTA during stable NREM sleep (top) and REM sleep (bottom). **c** Latency from NREM (top) or REM (bottom) sleep to wakefulness in each group. *n* = 5 mice in each group. **d** Representative EEG power spectrum and EEG-EMG trace data surrounding optogenetic stimulation during isoflurane-induced steady-state anesthesia in the ChR2 group. **e** Upper: EEG power of each 0.2 Hz bin from 0 to 25 Hz 60 s before and during light-stimulation in the ChR2 group. Lower: average EEG power of different frequency bands expressed as mean (gray for 60 s before light-stimulation and blue for during light-stimulation) ±SEM (shaded). **f** Quantitative of relative power of EEG signals across four bands (delta, theta, alpha, beta) before and during optogenetic stimulation in the ChR2 group. *n* = 10 trials from 5 mice. **g** Representative EEG power spectrum and EEG-EMG trace data surrounding optogenetic stimulation during isoflurane-induced steady-state anesthesia in the mCherry group.

**h** Upper: EEG power of each 0.2 Hz bin from 0 to 25 Hz 60 s before and during light-stimulation in the mCherry group. Lower: average EEG power of different frequency bands expressed as mean (gray for 60 s before light-stimulation and blue for during light-stimulation) ± SEM (shaded). **i** Quantitative of relative power of EEG signals across four bands before and during optogenetic stimulation in the mCherry group. *n* = 10 trials from 5 mice. **j** Representative EEG power spectrum and EMG trace data surrounding optogenetic stimulation during burst-suppression oscillations induced by deep anesthesia in the ChR2 group. **k** Quantification of burst duration before and during light stimulation in the ChR2 group. *n* = 10 trials from 5 mice. **l** Burst suppression ratio (BSR) before and during light stimulation in the ChR2 group. *n* = 10 trials from 5 mice. **m** Representative EEG power spectrum and EMG trace data surrounding optogenetic stimulation during burst-suppression oscillations induced by deep anesthesia in the mCherry group. **n** Quantification of burst duration before and during light stimulation in the mCherry group. *n* = 10 trials from 5 mice. **o** BSR before and during light stimulation in the mCherry group. *n* = 10 trials from 5 mice. Data are presented as mean ± SEM. Statistical comparisons were determined using Wilcoxon rank-sum test, two-way RMs ANOVA with Bonferroni *post hoc* test or paired Student's *t* test. *$P < 0.05$, **$P < 0.01$, ***$P < 0.001$.

A limited number of studies have explicitly investigated the activity rhythms of BNST GABAergic neurons across the natural sleep-wakefulness cycle. Here, using cell-type specific fiber photometry combined with electroencephalography (EEG), electromyography (EMG), and video recordings, we identified that BNST GABAergic neurons exhibited arousal-state-dependent alterations, with high activity in both wakefulness and REM sleep (Fig. 1). This is consistent with previous electrophysiological recordings demonstrating the majority of BNST neurons fire more frequently during wakefulness and REM sleep than during NREM sleep[24]. The similar phenomenon also occurs in wake-promoting GABAergic neuronal clusters of some other brain areas. For example, the GABAergic neurons in the ventral pallidal, dorsal raphe nucleus, and lateral hypothalamus that regulate wakefulness are active during both wakefulness and REM sleep[25–27]. This may be due to the mixed and functionally heterogeneous subpopulations of BNST GABAergic neurons. Single-cell resolution recordings utilizing optrodes or miniaturized microscope were required to further clarify the subpopulation responsible for wake promotion[28,29].

Sleep disturbances are detrimental to health and often entwined with stress and mood disorders[30]. Patients with posttraumatic stress disorder and compulsive behavior disorders commonly experience concurrent insomnia[31,32]. The BNST has been possibly overlooked in connecting neuropsychiatric illness and sleep architecture. It has been considered a crucial brain node in the response to stress and anxiety in both human and animal research. A brain functional magnetic resonance imaging study has revealed enhanced activity in the BNST in patients suffering from anxiety disorders[33]. In rodents, stress-induced hyper-anxiety is associated with increased BNST volumes[34]. Increased activity of BNST CRF cells, a subpopulation of GABAergic neurons and identified as stress-responsive cells, is highly associated with struggling bouts during restraint stress[35]. In our study, prolonged activation of BNST GABAergic neurons resulted in sustained wakefulness during a period of increased sleep drive (Supplementary Fig. 3). Based on the evidence above, we infer that excessive activation of the BNST possibly lead to hyper-arousal or insomnia in stress scenarios. Additionally, our findings suggest that chronic silencing of BNST GABAergic neurons increase both NREM and REM sleep (Fig. 5). Overall, BNST GABAergic neurons and their downstream neural circuits may be a potential target in the treatment of stress or psychiatry disorders associated insomnia.

The VTA is the major node of midbrain dopaminergic neurons involved in the sleep-wakefulness regulation and receives dense afferent fibers from the BNST[6,36]. Furthermore, an increased activity in the VTA was found following activation of BNST GABAergic neurons by micro-positron emission tomography[37]. Our study suggested that the VTA might act as a potential downstream pathway of the BNST in modulating wakefulness. According to Kudo et al.'s research, the BNST-VTA projection is predominantly GABAergic, with the downstream neurons in the VTA also being primarily GABAergic[21]. Therefore, its activation is predicted to disinhibit VTA dopaminergic neurons, ultimately leading to sleep to

wakefulness transition. In addition, BNST GABAergic neurons also send projections to other pivotal wake-promoting nuclei, including the paraventricular thalamus, lateral hypothalamus, and locus coeruleus[7,15]. Further investigation is required to determine whether these projection subpopulations are intermingled or distinct from those projecting to the VTA.

Sleep and GA exhibit many analogous neurophysiological characteristics. Firstly, both states exhibit functional decreases in body temperature, heart rate, respiratory rate, and blood pressure. Secondly, they share the common trait of reversible unconsciousness[38]. Lastly, the EEG oscillations induced by some anesthetic agents resemble those observed during sleep[39]. Recently, multiple lines of evidence have demonstrated that activation of wake-promoting brain areas could promote emergence and accelerate recovery of consciousness from GA[11,12,19]. Changes in the level of c-Fos expression after sleep deprivation or anesthesia suggest that BNST GABAergic neurons may be involved in the regulation of sleep-wakefulness and GA[17,18]. Here, in this study we also found that activation of the wake-promoting GABAergic neurons in the BNST induced EEG arousal from steady-state or deep anesthesia (Fig. 4). However, GA and sleep represent distinct states. GA is a neuropharmacological condition induced by drugs, while sleep is a natural physiological state[4]. Their molecular and neural circuit mechanisms may not be the same. In deep anesthesia, anesthetics can induce burst suppression, a pattern never observed during natural sleep[40]. Moreover, GA may contribute to drug-induced brain dysfunction and impair learning and memory. In contrast, sleep is important for memory processing and consolidation[41]. The integration of studies examining both physiological and anesthetic-induced loss of consciousness is emerging as a new research trend.

In summary, we identified BNST GABAergic neurons as a potential node in the regulation of emergence from natural sleep and anesthesia. These neurons exhibited arousal-state-dependent alterations, with high activity in both wakefulness and REM sleep. Our current study suggested that the BNST-VTA acts, at least in part, as a descending pathway in modulating physiological arousal and anesthesia emergence. Our research may contribute to unveiling the nature of consciousness, exploring more specific anesthetic drugs, and optimizing sleep quality.

## Methods
### Animals
*Vgat-Cre* mice were obtained from Jackson Laboratory. *Vgat-Cre* mice were crossed with Cre-dependent tdTomato reporter knock-in mice (B-tdTomato cKI mice, Beijing Biocytogen Inc.). 8 to 20-week-old mice of both sexes were utilized for the experiment. Mice at least 11 weeks old were used for recording and manipulation experiments. Mice were housed up to five per cage at a constant temperature (22 ± 1 °C) and humidity (50 ± 5%), with a standard 12-hour light-dark cycle (light turned on at 8:00 am). Cages were replaced twice a week, with water and food available ad libitum. Mice implanted with optical fiber were housed individually. All experimental

procedures were approved by the Third Military Medical University Animal Care and Use Committee and experiments were performed according to institutional animal welfare guidelines.

## Stereotaxic virus injections

Mice were induced with 2% isoflurane, and positioned in a stereotaxic frame (RWD Life Science Co., Ltd., Shenzhen, China) with a heated pad underneath. Constant depth of anesthesia was maintained using a facemask delivering 1.5% isoflurane. The skin was sanitized with iodine and locally anesthetized with 2% lidocaine. A vertical incision was made to reveal the skull, exposing bregma and lambda. Burr holes (0.5 mm × 0.5 mm) were drilled above the BNST.

For fiber photometry experiments, AAV-DIO-GCaMP6f (AAV2/9, titer: $0.5 \times 10^{12}$ vg mL$^{-1}$, Obio Biotechnology Co., Ltd. Shanghai, China, 60–80 nL) or AAV-DIO-EGFP (AAV2/9, titer: $0.5 \times 10^{12}$ vg mL$^{-1}$, Obio Biotechnology Co., Ltd. Shanghai, China, 60–80 nL) was unilaterally delivered into the BNST (AP: +0.14 mm, ML: 1.0 mm, DV: −4.20 mm) through a micropipette. For antidromic stimulation experiments, AAV-DIO-jRECO1a (AAV2/9; titer: $5.8 \times 10^{12}$ vg mL$^{-1}$, BrainCase Co., Ltd., 60–80 nL) and AAV-DIO-hChR2-EGFP (AAV2/9; titer: $5.4 \times 10^{12}$ vg mL$^{-1}$, Taitool Bioscience Co., 60–80 nL) were unilaterally injected into the BNST. For optogenetic experiments, AAV-DIO-ChR2-mCherry (AAV2/9; titer: $1.2 \times 10^{13}$ vg mL$^{-1}$, Taitool Bioscience Co.; 60–80 nL on each side) was injected bilaterally into the BNST. For ablation experiments, AAV-DIO-taCaspase3 (AAV2/9; titer: $1.6 \times 10^{13}$ vg mL$^{-1}$, Taitool Bioscience; 60–80 nL on each side) was bilaterally injected into the BNST. For the control group, AAV-DIO-mCherry (AAV2/9; titer: $1.2 \times 10^{13}$ vg mL$^{-1}$, Taitool Bioscience Co.; 60–80 nL on each side) was used. The glass micropipette remained at the injection site for approximately 5 min, and was removed slowly and carefully. After surgery, meloxicam (1 mg kg$^{-1}$, Boehringer Ingelheim, Germany) was given for 3 days. The mice were given at least three weeks to recover prior to the second operation.

## EEG-EMG signals recording and analysis

To capture EEG activity, three silver-plated EEG electrodes were implanted in three tiny holes drilled above the cortex, with two above the frontal lobe and the third above the parietal lobe as a reference electrode[19,42]. Silver-plated EMG electrodes were then inserted separately between the neck muscle tissue to measure EMG activity. The EEG-EMG device was fixed to the skull using dental cement. Meloxicam (1 mg kg$^{-1}$, Boehringer Ingelheim, Germany) was given for 3 days after surgery. Signals were measured through electrophysiological recording cables 7 days after EEG and EMG surgery.

All recorded EEG-EMG signals were filtered by a band-pass filter (EEG, 0–25 Hz; EMG, 10–70 Hz). Analysis software (SleepSign for animals, Kissei Comtec, Japan) was used to automatically analyze the sleep-wakefulness states[43]. EEG signals were divided into non-overlapping 4 s epochs for analysis. NREM sleep was characterized by high amplitude in EEG activity, dominated by delta waves, and minimal EMG amplitude. REM sleep was dominated by theta waves, with a further decline in EEG amplitude and a lack of tonic EMG activity. Wakefulness was defined as desynchronized low amplitude EEG and high EMG activity. The results were inspected manually and corrected when appropriate. A state transition was identified once the EEG-EMG criteria change was apparent for longer than 50% of the epoch duration.

## Fiber photometry

Fiber photometry system was used for Ca$^{2+}$ signals recording[42,44,45]. The optical fiber (0.2 mm diameter, NA 0.48, Doric lenses, Quebec City, QC, Canada) was fastened by a short steel cannula. The fiber tip was extended approximately 5 mm outside of the cannula and slowly implanted 50 μm above the virus injection region. Mice were given at least 7 days to recover prior to Ca$^{2+}$ signals recording. To observe the activities of BNST GABAergic neurons during the sleep-wakefulness cycle, the optical fiber was connected through a PC adapter to a 473 nm blue laser diode fiber photometric system (FOM-02M Fiber OptoMeter, Suzhou Institute of

Biomedical Engineering and Technology, China). The light intensity was 0.22 mW/mm$^2$ at the fiber tip. The collected Ca$^{2+}$ signals were digitized using customized acquisition LabVIEW software (National instrument) with a sampling frequency of 2000 Hz. The derived photometry data were low-pass filtered using a Savitzky–Golay FIR smoothing filter with 50 side points and a third order polynomial. The Ca$^{2+}$ transients were then calculated by $\Delta F/F = (F - F_{baseline})/F_{baseline}$, where $F_{baseline}$ is collected during the current recording period. The EEG-EMG signals (200 Hz) and behavioral videos (30 Hz) were simultaneously recorded throughout sleep-wakefulness cycles. Offline event makers were used to synchronize all these signals. The sleep-wakefulness state was identified through synchronous EEG-EMG signals. To quantify Ca$^{2+}$ activity during the sleep-wakefulness cycle, the Ca$^{2+}$ signals were averaged over the duration of wake, NREM or REM sleep episodes. For state transition analysis, the Ca$^{2+}$ signals averaged 30 s preceding the state transition were compared to those 30 s after the state transition. For antidromic stimulation experiments, the optical fiber was linked to the fiber photometry system (Inper Technology, China), utilizing a 561 nm laser to elicit jRECO1a fluorescence signals.

## Optogenetics

Upon bilateral optical fiber fixation, recording experiments were conducted after a recovery of at least 7 days. EEG-EMG signals and behavioral videos were continuously observed. When mice entered stable NREM or REM sleep (at least 15 - 20 s), 473 nm laser pulses (5 mW at the fiber tip, 10 ms, 20 Hz) were delivered for 20 s in each trial. The inter-trial intervals were 5–10 min. The laser pulse control was carried out using customized software on the LabVIEW platform[42,44]. For prolonged stimulation, mice were allowed to adapt for one hour before recording. The programmed blue light-pulse trains (10 ms, 20 Hz for 20 s, every 60 s for 1 h) lasted between 10:00 am to 11:00 am. For optical stimulation of BNST GABAergic terminals, the fiber was implanted in the VTA (AP: 3.1 mm, ML: 0.4 mm, DV: −4.20 mm). To investigate the potential antidromic stimulation through photostimulation of axon terminals in the VTA, 473 nm laser pulses (1 mW, 5 mW, 8 mW and 10 mW at the fiber tip, 10 ms duration, 20 Hz) were delivered for 10 s in each trial. And Ca$^{2+}$ recordings were conducted under 0.6–0.8% isoflurane anesthesia with 1 min inter-trial intervals. Light power intensity at the fiber tip was measured by a power meter (PM100D, Thorlabs).

## GA experiments

Mice were placed in an acrylic glass barrel linked to an anesthesia monitor (IntelliVue MP60, Royal Dutch Philips Electronics, Netherlands) to detect the concentration of isoflurane. After acclimation to the experimental environment, Ca$^{2+}$ signals were recorded for 20 min prior to administration of inhaled anesthetics. 1.4% isoflurane (oxygen flow rate: 2 L/min) or only oxygen was continuously delivered for the following 20 min. After the vaporizer was switched off, it took 10 min for the isoflurane concentration to drop to 0%, as monitored by MP60. Once the isoflurane was washed out, the recording continued for another 10 min indicating recovery. The total recording time was classified into three portions, including 20 min before isoflurane anesthesia (Pre-Iso), 30 min during isoflurane anesthesia (During-Iso, from isoflurane vaporizer turning on to isoflurane washing out), and 10 min after isoflurane anesthesia (Post-Iso). The Ca$^{2+}$ signals were downsampled to 200 Hz and zeroed by calculating the median value of 20 min before isoflurane administration and subtracting this from the GCaMP6s signals throughout the recording period[27]. To quantify the change in Ca$^{2+}$ activity across different states, the z-score transformed $\Delta F/F$ was normalized using the standard deviation (SD) of the Ca$^{2+}$ signals 20 min before isoflurane administration and averaged over the duration of different states[46].

For optogenetic activation experiments during anesthesia, mice were maintained at a steady-state using 0.8% isoflurane or burst suppression mode using 1.2% isoflurane for 10−15 min. 473 nm laser pulses (5 mW at the fiber tip, 10 ms, 20 Hz) were delivered for 60 s in each trial. EEG-EMG signals and video recordings were performed at the same time. To obtain

EEG power spectrum, the original EEG data were band-pass filtered (0–25 Hz) and calculated using fast-Fourier transformation (FFT) method. The EEG frequency band was defined by four bands (delta: 0.5–4 Hz, theta: 4–10 Hz, alpha: 10–15 Hz, beta: 15–25 Hz). The EEG power of each 0.2 Hz bin was shown and the relative power of EEG signals across four bands was calculated. BSR was used for quantifying burst suppression. Raw EEG data were processed using a Hilbert transform by MATLAB R2020a software, and a visually-based voltage threshold was established for each mouse. Binary series was utilized for BSR calculation. If the EEG amplitude exceeded the threshold, EEG signals were transformed to a value of 1, otherwise, a value of 0 was assigned. In the present study, the minimum duration of a suppression period was set to 0.5 s, and BSR was calculated as the percentage of time spent in suppression per minute.

### Arousal scoring

The experimenter conducting the assessment was blind to the intervention to prevent bias. Animals were observed for signs of arousal for 60 s after the onset of optogenetic stimulation during steady-state anesthesia and were scored using the method of Taylor et al.[47]. Spontaneous movements of the leg, head, and whisker were scored as one of three levels as 0 (absent), 1 (mild), or 2 (moderate) in intensity. For example, during optogenetic stimulation, leg movement during light stimulation was scored as 0 if the mouse remained motionless in a prone position, 1 if the mouse struggled with its leg, and 2 if the mouse touched the ground with all four paws. Orienting was scored as 0 if they remained prone, and 2 if they recovered its righting reflex. Furthermore, walking during light stimulation was scored as 0 if they remained motionless in a prone position, 1 if they struggled with its leg, and 2 if the mouse touched the ground with all four paws. The maximum score for each mouse is 10.

### Immunohistochemistry

After being deeply anesthetized with sodium pentobarbital (10 mg/kg, i.p.), mice were transcardially perfused using 4 °C saline, followed by 4% paraformaldehyde (PFA) at 4 °C. Brains were carefully extracted using forceps and post-fixed in 4% PFA. Coronal 50 μm thick brain sections containing the BNST or VTA were collected using a cryostat (NX50, Thermo). Brain slices were washed three times with PBS solution for 10 min each to remove residual fixative compound. Following permeabilization, the slices were blocked during 2 h incubation in PBS containing 10% normal donkey serum and 0.3% Triton X−100. For NeuN staining, brain slices were incubated with rabbit anti-NeuN (1:400, ab177487, Abcam) for 48 h at 4 °C. Afterward, the sections were washed three times for 10 min and incubated with a secondary antibody Alexa Fluor 488 donkey anti-rabbit (1:400, AB_2556546, Invitrogen) for 2 h at room temperature. Sections were stained with DAPI (Beyotime, Shanghai, China, C1006) at 1:2500 for 15 min and washed three times for ten min. The sections were mounted on slides, air dried, sealed with anti-quenching fluorescent sealer, and covered for imaging with a scanning confocal microscope (TCS SP5, Leica).

### In situ hybridization

*Vgat-Cre* transgenic mice were deeply anesthetized with 1% pentobarbital sodium. Then they were transcardially perfused with saline followed by 4% PFA in PBS. Brains were obtained, post-fixed in 4% PFA for 24 h and dehydrated in sucrose solution (10%, 20%, and 30%) until sectioning. Coronal brain slices were cut at 15 μm using a cryostat and were stained using RNAscope Multiplex Fluorescent Assays V2 according to the manufacturer's instructions (Advanced Cell Diagnostics, USA). Brain slices were hybridized with the mixed target probes (*RNAscope Probe Gad1, RNAscope Probe Gad2, RNAscope Probe GFP*), followed by amplification. Sections were stained with DAPI and mounted with Pro Long Gold antifade mountant. Images were acquired with a scanning confocal microscope (TCS SP5, Leica).

### In vitro electrophysiological recordings

Electrophysiological recordings were performed as previously described[27,48]. Mice were anesthetized with isoflurane and intracardially perfused with an ice-cold cutting solution containing (in mM): 92 NMDG, 2.5 KCl, 1.25 NaH$_2$PO$_4$, 30 NaHCO$_3$, 20 HEPES, 25 glucose, 2 thiourea, 5 Na-ascorbate, 3 Na-pyruvate, 0.5 CaCl$_2$, and 10 MgSO$_4$. Coronal brain slices containing BNST (300 μm thickness) were obtained in ice-cold cutting solution bubbled with 95% O$_2$ and 5% CO$_2$ with a Vibroslicer (752 M; Campden Instruments). Slices were recovered in an interface chamber with artificial cerebrospinal fluid (ACSF) containing (in mM): 124 NaCl, 2.5 KCl, 1.25 NaH$_2$PO$_4$, 24 NaHCO$_3$, 12.5 glucose, 5 HEPES, 2 CaCl$_2$, and 2 MgSO$_4$, bubbled with carbogen at 34 °C for 45 min and kept at room temperature until recordings started. Slices were then transferred to recording chamber and continuously perfused with ACSF at the rate of 1–3 ml per min. Slices were visualized on an upright fixed stage microscope which equipped with epifluorescence and infrared-differential interference contrast illumination to identify fluorescently tagged cells. During recording, the patch electrodes were fabricated from thick-walled borosilicate glass capillaries and had resistances between 3 and 6 MΩ when filled with an intracellular solution containing (in mM) 145 K-Gluconate, 10 HEPES, 1 EGTA, 2 Mg-ATP, 0.3 Na$_2$-GTP, and 2 MgCl$_2$. To verify the functional expression of ChR2-mCherry, we used cell-attached patch-clamp recordings in brain slices. The BNST slices were stimulated by 473 nm laser pulses (5 mW at the fiber tip, 10 ms, 20 Hz for 3 s) delivered via an optical fiber.

In order to investigate the direct effects of isoflurane on GABAergic neurons in the BNST, whole-cell patch clamp recordings were performed and action potential (AP) were recorded in current-clamp mode. *Vgat-tdTomato* mice were applied in this experiment. To evoke APs from GABAergic neurons in the BNST, positive current (5 - 15 pA) was injected into the cell under current-clamp mode to induce a steady firing activity, isoflurane was bath applied. The clinically relevant isoflurane solution and the concentration were prepared and calculated as previously reported[49]. Data were acquired with Axon 700B Amplifier (Molecular Devices) and further analyzed with pCLAMP 9.2 software (Molecular Devices).

### Statistics and reproducibility

Statistical analyses were conducted using Graph Pad Prism 9.0 (GraphPad Software, USA) and SPSS 27.0 software (IBM SPSS Statistics for Windows, USA). Normality tests were applied before analysis. Paired or unpaired Student's *t*-tests were performed for comparisons between the two groups. One-way or two-way RMs ANOVA was used for multigroup comparison, followed by Bonferroni *post hoc* comparison tests. If data were not normally distributed, non-parametric tests (Wilcoxon rank-sum test) were used for two independent groups. Data in this study were represented as mean ± standard error of the mean (SEM). In all cases, two-tailed $P < 0.05$ was considered statistically significant. Statistical details of experiments were found in the figure legends.

### Reporting summary

Further information on research design is available in the Nature Portfolio Reporting Summary linked to this article.

### Data availability

The source data supporting the depicted graphs and charts have been included in the Supplementary Data 1. Further data are available from the corresponding author on reasonable request.

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

## Acknowledgements
This study was supported by grants from the National Natural Science Foundation of China (82101282, 82001398, 31925018, 32171096, 32200838), the Natural Science Foundation Project of Chongqing (CSTB2022NSCQ-MSX1137).

## Author contributions
X.C. and J.X. conceived the project. X.C., J.X., H.Liu and H.Q. designed the experiments; M.L., J.X., M.G., and H.Li performed the experiments; X.C., J.X., H.Q., and H.Liu devised the data analysis methods; J.X., S.L., X.L. and W.L. performed the data analysis; J.X., H.Liu and X.C. inspected the data and evaluated the findings; J.X., M.L., X.C., H.Q. and H.Liu wrote the manuscript with the help from all authors.

## Competing interests
The authors declare no competing interests.
