## [Peer review file · Communications Biology]

Reviewers' comments:

Reviewer #1 (Remarks to the Author):

Li et al. present a manuscript describing that the bed nucleus of the stria terminalis (BNST) GABAergic neurons is an important neural substrate for arousal from sleep and anesthesia.

The experiments are mostly novel, and various modern techniques were used to query these neurons' physiology and behavioral function. The authors put together a manuscript with many interesting features. Their main findings are:

1. BNST GABAergic neurons exhibit arousal-state-dependent alterations, with high activities in both wakefulness and REM sleep, but suppressed during anesthesia.
2. Optogenetic activation of these neurons can initiate and maintain wakefulness and even induce arousal from anesthesia.
3. BNST GABAergic →VTA projections may be involved in such arousal function.

This reviewer believed it is suitable for publication in Brain Communications after the proper address of these concerns:

1. There is a debate about whether sleep and anesthesia share the same neural substrate. Thus, as a manuscript puts sleep and anesthesia together, there should be more discussion about their similarity and differences.
2. There is a possibility that terminal stimulation in VTA produces antidromic spikes in BNST cell bodies. The author should exclude such a possibility.

Reviewer #2 (Remarks to the Author):

Li and colleagues investigated the role of GABAergic neurons in the bed nucleus of the stria terminalis (BNST) in sleep and general anesthesia. They utilized fiber photometry and optogenetic approaches to specifically manipulate BNST GABAergic neurons in Vgat-cre mice to test the sleep-wake transition and EEG alterations under isoflurane general anesthesia.

The authors observed that activation of the BNST GABAergic neurons significantly promoted wake initiation and maintenance, as well reanimating from anesthesia. Moreover, optogenetic results verified that BNST GABA-ventral tegmental area (VTA) neural circuit mediated the wake promotion effect, rather than recovery from general anesthesia.

The experiments are basically well designed and carried out. The conclusion was supported by the data. However, I have some concerns.

Major comments:

1. Although previous study has revealed that the BNST GABAergic neurons could trigger immediate transition from NREM sleep to wakefulness (Shota et al., J Neurosci 2017), the wake promoting effect of BNST GABAergic neurons in this article seems to be too significant, as both acute and prolonged optical activation could induce over 90% wakefulness. To further confirm the results, you'd better apply optical inhibition and chemogenetics, and carefully discuss about the underlying meaning of the intense wake promoting effect, which may play an important role in stress or psychiatry disorders associated insomnia.
2. Different from Shota et al. study, the authors found activation of BNST GABAergic neurons could

also induce immediate transition from REM sleep to wakefulness. However, in the EEG spectrum, the REM sleep is not typical enough, which may largely influence the reliability of results. The authors should carefully check associated data and results.

3. The article investigated the effect of BNST GABAergic neurons on general anesthesia, however, there were no description about the behavioral changes.

4. From Figure 1e, the quantification of calcium activity during wakefulness seems inconsistent with the data from figure 1f.

5. You'd better verify that only GABAergic neurons in the BNST were manipulated in this study by using immunohistochemistry double-label staining, and to ensure if the AAV-GCaMP6f and AAV-ChR2 work by immunohistochemistry staining in vivo or whole-cell patch-clamp recordings in vitro.

Minor comments:

1. Mouse gender used for experiments should be stated in the methods. In addition, 8 week old mice used in the study is still in puberty.

2. After surgery, if there was any analgesics for the mice?

3. Please note the superscript for calcium, as on line 348.

Reviewer #3 (Remarks to the Author):

The neural circuits through which anesthetics suppress consciousness are not yet fully understood. In this study, Li et al. investigated the role of the bed nucleus of the stria terminalis (BNST) GABAergic neurons in isoflurane-induced anesthesia. Using fiber photometry in combination with EEG/EMG recordings to assess the state of consciousness, they found that BNST Ca activity decreased during anesthesia and increased again after isoflurane termination. They also used optogenetics targeting GABAergic BNST neurons to probe the emergence from anesthesia. One key finding is that BNST GABAergic projections to ventral tegmental area (VTA) neurons, likely GABA neurons, have prominent contributions to the control of anesthesia and wakefulness.

Many brain areas/nuclei are involved in the effects of anesthesia. The hypnotic role of VTA GABAergic neurons in sleep-wake regulation and anesthesia has been described in a number of articles published in *Nat. Neurosci*, *eLife*, and *J. Neurosci.*, and inhibition or reactivation of BNST neurons upstream of the VTA can apparently facilitate recovery from unconsciousness via VTA neurons, likely among many other brain areas. I find it difficult to see the conceptual advance of this paper and wonder whether papers like this one really bring us much closer to a comprehensive understanding of anesthesia.

The work does not show the extent to which BNST neurons facilitate recovery of consciousness compared with other brain regions, such as the dopaminergic VTA neurons, which are downstream of BNST neurons and are also likely to be allosterically inhibited by isoflurane during anesthesia and may promote arousal from anesthesia upon reactivation.

In addition, I have the following specific comments:

- The photometry data show that BNST neurons are primarily active during REM sleep and that activity actually decreases from REM sleep to wakefulness. However, optogenetic activation of these neurons induces wakefulness from NREM or REM sleep. What happens when these neurons are

chemogenetically activated?

- I see no evidence for the physiological importance of BNST neurons for arousal, anesthesia and recovery from anesthesia by performing experiments on loss of function of BNST neurons, e.g. by ablation or inhibition of BNST neurons.

- This work would benefit from evidence that isoflurane acts on BNST neurons. How is isoflurane-induced anesthesia, and possibly BNST calcium activity, affected by loss of GABAA receptor function in the BNST? Can Isoflurane deactivate BNST GABergic neurons, e.g. in electrophysiology studies?

We would like to express our deep appreciation to the reviewers for their constructive comments on our manuscript (COMMSBIO-23-1049). In order to address each of their comments, we made the following changes. We have marked the major changes in the manuscript in red.

- (1) We have added a paragraph in the introduction to introduce the roles of GABAergic neurons in the brain on sleep-wakefulness and general anesthesia regulations.
- (2) We have standardized the range of the y-axis in Fig. 1f.
- (3) We have added new data for *in situ* hybridization to verify the labeled cells were GABAergic neurons in Supplementary Fig. 1a-b.
- (4) We have performed a new control experiment to exclude motion artifacts in the Ca²⁺ recording experiments in Supplementary Fig. 1c-f.
- (5) We have added new data to investigate the direct effects of isoflurane on BNST GABAergic neurons through electrophysiology studies in Supplementary Fig. 2.
- (6) We have added new data to verify the efficiency of ChR2 in Fig. 3c and changed more typical examples in Fig. 3d and Fig. 6b.
- (7) We have added new data for ablation (Fig. 5) of BNST GABAergic neurons during natural sleep and general anesthesia.
- (8) We have analyzed the arousal scores during steady-state anesthesia (Supplementary Fig. 4, Supplementary Table 1, 2) with photostimulations on BNST GABAergic neurons and BNST-VTA projection.
- (9) We have performed a new control experiment to demonstrate that light stimulation of 5 mW did not induce antidromic spikes in Supplementary Fig. 5.
- (10) We have discussed similarities and differences between sleep and anesthesia in the discussion part (lines 284-289 and 293-299).
- (11) We have provided detailed methodology for animals, arousal scoring, *in situ* hybridization, and *in vitro* electrophysiological recordings.

Reviewer Comments:

Reviewer #1 (Remarks to the Author):

Li et al. present a manuscript describing that the bed nucleus of the stria terminalis (BNST) GABAergic neurons is an important neural substrate for arousal from sleep and anesthesia. The experiments are mostly novel, and various modern techniques were used to query these neurons' physiology and behavioral function. The authors put together a manuscript with many interesting features. Their main findings are:

1. BNST GABAergic neurons exhibit arousal-state-dependent alterations, with high activities in both wakefulness and REM sleep, but suppressed during anesthesia.
2. Optogenetic activation of these neurons can initiate and maintain wakefulness and even induce arousal from anesthesia.
3. BNST GABAergic →VTA projections may be involved in such arousal function.

This reviewer believed it is suitable for publication in Brain Communications after the proper address of these concerns:

We thank the reviewer very much for the appreciation of our work and the constructive comments that are essential for improving our manuscript.

1. There is a debate about whether sleep and anesthesia share the same neural substrate. Thus, as a manuscript puts sleep and anesthesia together, there should be more discussion about their similarity and differences.

This is a very important suggestion. Following this suggestion, we now added this discussion (lines 284-289 and 293-299).

2. There is a possibility that terminal stimulation in VTA produces antidromic spikes in BNST cell bodies. The author should exclude such a possibility.

Thanks a lot for this suggestion. We have now added a control experiment, demonstrating that light stimulation of 5 mW or less did not induce antidromic spikes (Supplementary Fig. 5). We used light intensity of 5 mW for axonal stimulation in the experiments.

Reviewer #2 (Remarks to the Author):

Li and colleagues investigated the role of GABAergic neurons in the bed nucleus of the stria terminalis (BNST) in sleep and general anesthesia. They utilized fiber photometry and optogenetic approaches to specifically manipulate BNST GABAergic neurons in Vgat-cre mice to test the sleep-wake transition and EEG alterations under isoflurane general anesthesia.

The authors observed that activation of the BNST GABAergic neurons significantly promoted wake initiation and maintenance, as well reanimating from anesthesia. Moreover, optogenetic results verified that BNST GABA- ventral tegmental area (VTA) neural circuit mediated the wake promotion effect, rather than recovery from general anesthesia.

The experiments are basically well designed and carried out. The conclusion was supported by the data. However, I have some concerns.

We would like to express our gratitude to the reviewer for recognizing the value of our work and the thoughtful feedback and suggestions that are indispensable for the enhancement of our manuscript.

Major comments:

1. Although previous study has revealed that the BNST GABAergic neurons could trigger immediate transition from NREM sleep to wakefulness (Shota et al., J Neurosci 2017), the wake promoting effect of BNST GABAergic neurons in this article seems to be too significant, as both acute and prolonged optical activation could induce over 90% wakefulness. To further confirm the results, you'd better apply optical inhibition and chemogenetics, and carefully discuss about the underlying meaning of the intense wake promoting effect, which may play an important role in stress or psychiatry disorders associated insomnia.

Following your suggestion and that of another reviewer, we conducted a silencing experiment using cell-type specific taCaspase3 to ablate the BNST GABAergic neurons. We confirmed that ablation of BNST GABAergic neurons decreased the amount of wakefulness and increased the amount of NREM and REM sleep during the dark phase (Fig. 5a-f). Our results showed a more significant wake promoting effect of BNST GABAergic neurons. The potential explanation may lie in the use of distinct transgenic animals. The Vgat-Cre mice are a preferable choice for investigating the function of

GABAergic neurons compared to Gad67-Cre mice. According to previous studies, Gad67-Cre transgenic animals might exhibit non-specific labeling of GABAergic neurons (Mickelsen et al., Nat Neurosci, 2019; Judd et al., Elife, 2021). And we also carefully discussed about the underlying meaning of the intense wake promoting effect in stress and psychiatry disorders (lines 258-272).

2. Different from Shota et al. study, the authors found activation of BNST GABAergic neurons could also induce immediate transition from REM sleep to wakefulness. However, in the EEG spectrum, the REM sleep is not typical enough, which may largely influence the reliability of results. The authors should carefully check associated data and results. Following the reviewer's suggestion, we have changed a more typical example in Fig. 3d. We also carefully reviewed associated data and results to ensure the validity and robustness of our findings.

3. The article investigated the effect of BNST GABAergic neurons on general anesthesia, however, there were no description about the behavioral changes.

Following the reviewer's suggestion, we have added behavioral changes in Supplementary Fig. 4, Supplementary Table 1, 2.

4. From Figure 1e, the quantification of calcium activity during wakefulness seems inconsistent with the data from figure 1f.

Apologies for any confusion. We have now standardized the range of y-axis in Fig. 1f to match that of Fig. 1e.

5. You'd better verify that only GABAergic neurons in the BNST were manipulated in this study by using immunohistochemistry double-label staining, and to ensure if the AAV-GCaMP6f and AAV-ChR2 work by immunohistochemistry staining in vivo or whole-cell patch-clamp recordings in vitro.

Following the reviewer's suggestion, we have conducted *in situ* hybridization experiment to confirm that the labeled cells in the BNST were GABAergic neurons (Supplementary Fig. 1b). We have also conducted a control experiment by recording EGFP-labeled BNST GABAergic neurons to exclude motion artifacts in the Ca²⁺ recording experiments (Supplementary Fig. 1c-f). In addition, we have included a set of cell-attached recording data to verify the efficiency of AAV-ChR2 (Fig. 3c).

Minor comments:

1. Mouse gender used for experiments should be stated in the methods. In addition, 8 week old mice used in the study is still in puberty.

8 to 20-week-old mice of both sexes were utilized for the experiment. Mice aged 8-week or older were used for virus injection. And 3-4 weeks after virus expression, mice (at least 11 weeks old) were used for recording and manipulation experiments. Following this suggestion, now we have explained these details in the Methods part.

2. After surgery, if there was any analgesics for the mice?

Yes, after the surgery, analgesics (meloxicam, 1 mg kg⁻¹, Boehringer Ingelheim, Germany) were administered to the mice to manage pain and ensure their well-being.

3. Please note the superscript for calcium, as on line 348.

Sorry for this mistake, and now we have corrected it.

Reviewer #3 (Remarks to the Author):

The neural circuits through which anesthetics suppress consciousness are not yet fully understood. In this study, Li et al. investigated the role of the bed nucleus of the stria terminalis (BNST) GABAergic neurons in isoflurane-induced anesthesia. Using fiber photometry in combination with EEG/EMG recordings to assess the state of consciousness, they found that BNST Ca activity decreased during anesthesia and increased again after isoflurane termination. They also used optogenetics targeting GABAergic BNST neurons to probe the emergence from anesthesia. One key finding is that BNST GABAergic projections to ventral tegmental area (VTA) neurons, likely GABA neurons, have prominent contributions to the control of anesthesia and wakefulness.

Many brain areas/nuclei are involved in the effects of anesthesia. The hypnotic role of VTA GABAergic neurons in sleep-wake regulation and anesthesia has been described in a number of articles published in *Nat. Neurosci.*, *eLife*, and *J. Neurosci.*, and inhibition or reactivation of BNST neurons upstream of the VTA can apparently facilitate recovery from unconsciousness via VTA neurons, likely among many other brain areas. I find it difficult to see the conceptual advance of this paper and wonder whether papers like this one really bring us much closer to a comprehensive understanding of anesthesia.

In recent years, an increasing number of neural substrates associated with inhaled induction and emergence have been identified (Bao, et al., *Curr Biol*, 2023). Nevertheless, the precise involvement of the BNST in the regulation of sleep-wakefulness and anesthesia remains to be elucidated.

The neural circuits that regulate inhaled GA induction and emergence (Bao, et al., *Curr Biol*, 2023).

For the first time, we recorded the calcium activities of BNST GABAergic neurons during sleep-wakefulness cycles in freely moving mice, and also documented their calcium activity under anesthesia. Secondly, previous studies only reported the wake-promoting effect of BNST GABAergic neurons from natural sleep. Here we found that optogenetic activation of BNST GABAergic neurons induced arousal from isoflurane anesthesia. Finally, we also studied the downstream target of these BNST GABAergic neurons on the wake promoting function from natural sleep and general anesthesia, and found this function was partially mediated by BNST-VTA pathway.

The work does not show the extent to which BNST neurons facilitate recovery of consciousness compared with other brain regions, such as the dopaminergic VTA neurons,

which are downstream of BNST neurons and are also likely to be allosterically inhibited by isoflurane during anesthesia and may promote arousal from anesthesia upon reactivation. Kudo et al. reported that the BNST-VTA projection is mainly GABAergic, and the downstream neurons in VTA are mainly GABAergic. Therefore, the activation of BNST-VTA projection is predicted to disinhibit VTA dopaminergic neurons (Kudo et al., *J Neurosci*, 2012). In future studies, we will compare the extent in facilitating recovery of consciousness of BNST GABAergic neurons with that of VTA dopaminergic neurons. However, in this study, we focus on the functions of the BNST GABAergic neurons and their projections to the VTA in the arousal control from natural sleep and anesthesia.

In addition, I have the following specific comments:

- The photometry data show that BNST neurons are primarily active during REM sleep and that activity actually decreases from REM sleep to wakefulness. However, optogenetic activation of these neurons induces wakefulness from NREM or REM sleep. What happens when these neurons are chemogenetically activated?

Kodani et al. have found that chemogenetic activation of BNST GABAergic neurons induced an increase in wakefulness, and a decrease in both NREM sleep and REM sleep in 2 h after CNO injection (Kodani et al., *J Neurosci*, 2017). Together with our optogenetic activation data, these results suggest that activation of BNST GABAergic neurons promotes wakefulness, not REM sleep. The similar phenomenon also exists in other brain regions. The GABAergic neurons in the ventral pallidum, dorsal raphe nucleus, and lateral hypothalamic are both wakefulness and REM sleep active (Li et al., *Mol Psychiatry*, 2021; Cai et al., *Sleep*, 2022; Venner et al., *Curr Biol*, 2016). Yet, chemogenetic activation of these GABAergic neurons could promote wakefulness while diminishing NREM and REM sleep.

- I see no evidence for the physiological importance of BNST neurons for arousal, anesthesia and recovery from anesthesia by performing experiments on loss of function of BNST neurons, e.g. by ablation or inhibition of BNST neurons.

Following this suggestion, we have now added these loss of function experiments in BNST GABAergic neurons. Ablation of these neurons decreased the amount of wakefulness and increased the amount of NREM sleep and REM sleep during the dark phase (Fig. 5a-f, see the effect in induction or emergence of anesthesia in Fig. 5g-h).

- This work would benefit from evidence that isoflurane acts on BNST neurons. How is isoflurane-induced anesthesia, and possibly BNST calcium activity, affected by loss of GABA_A receptor function in the BNST? Can Isoflurane deactivate BNST GABAergic neurons, e.g. in electrophysiology studies?

Yes, we have now provided new electrophysiological data showing that isoflurane deactivated the firing rate of BNST GABAergic neurons in Supplementary Fig. 2. Multiple cellular and molecular mechanisms have been previously found for isoflurane-induced anesthesia, including reduction in neuronal excitability, impairment of action potential conduction, inhibition of Ca²⁺ influx, and suppression of synaptic vesicle exocytosis. (see the review Platholi et al., *Curr Neuropharmacol*, 2022). In particular, GABA_A receptor has been found to be a potential anesthetic target. Isoflurane can directly activate GABA_A receptors by potentiating GABA-induced Cl⁻ currents (Franks, *Nat Rev Neurosci*, 2008; Hu et al., *Nat Neurosci*, 2023). Previous studies have reported that BNST GABAergic neurons

express GABA_A receptors (Dumont et al., J Neurosci, 2004; Romaguera et al., Cell Reports, 2020). Thus, isoflurane may activate GABA_A receptors in BNST GABAergic neurons, and reduce their firing rates and Ca²⁺ activities. In addition, isoflurane can also reduce the excitability and synaptic neurotransmission of BNST GABAergic neurons by other specific mechanism, such as the inhibition of sodium currents (Zhao et al., Anesthesiology, 2019). In the future studies, we will study the detailed mechanism of isoflurane anesthesia on BNST neurons.

REVIEWERS' COMMENTS:

Reviewer #1 (Remarks to the Author):

The authors have thoroughly addressed my previous concerns. I am confident that this manuscript is ready for acceptance.

Reviewer #2 (Remarks to the Author):

No comments.

We would like to express our deep appreciation to the reviewers for their constructive comments on our manuscript (COMMSBIO-23-1049). In order to address each of their comments, we made the following changes. We have marked the major changes in the manuscript in red.

- (1) We have added a paragraph in the introduction to introduce the roles of GABAergic neurons in the brain on sleep-wakefulness and general anesthesia regulations.
- (2) We have standardized the range of the y-axis in Fig. 1f.
- (3) We have added new data for *in situ* hybridization to verify the labeled cells were GABAergic neurons in Supplementary Fig. 1a-b.
- (4) We have performed a new control experiment to exclude motion artifacts in the Ca²⁺ recording experiments in Supplementary Fig. 1c-f.
- (5) We have added new data to investigate the direct effects of isoflurane on BNST GABAergic neurons through electrophysiology studies in Supplementary Fig. 2.
- (6) We have added new data to verify the efficiency of ChR2 in Fig. 3c and changed more typical examples in Fig. 3d and Fig. 6b.
- (7) We have added new data for ablation (Fig. 5) of BNST GABAergic neurons during natural sleep and general anesthesia.
- (8) We have analyzed the arousal scores during steady-state anesthesia (Supplementary Fig. 4, Supplementary Table 1, 2) with photostimulations on BNST GABAergic neurons and BNST-VTA projection.
- (9) We have performed a new control experiment to demonstrate that light stimulation of 5 mW did not induce antidromic spikes in Supplementary Fig. 5.
- (10) We have discussed similarities and differences between sleep and anesthesia in the discussion part (lines 284-289 and 293-299).
- (11) We have provided detailed methodology for animals, arousal scoring, *in situ* hybridization, and *in vitro* electrophysiological recordings.

Reviewer Comments:

Reviewer #1 (Remarks to the Author):

Li et al. present a manuscript describing that the bed nucleus of the stria terminalis (BNST) GABAergic neurons is an important neural substrate for arousal from sleep and anesthesia. The experiments are mostly novel, and various modern techniques were used to query these neurons' physiology and behavioral function. The authors put together a manuscript with many interesting features. Their main findings are:

1. BNST GABAergic neurons exhibit arousal-state-dependent alterations, with high activities in both wakefulness and REM sleep, but suppressed during anesthesia.
2. Optogenetic activation of these neurons can initiate and maintain wakefulness and even induce arousal from anesthesia.
3. BNST GABAergic →VTA projections may be involved in such arousal function.

This reviewer believed it is suitable for publication in Brain Communications after the proper address of these concerns:

We thank the reviewer very much for the appreciation of our work and the constructive comments that are essential for improving our manuscript.

1. There is a debate about whether sleep and anesthesia share the same neural substrate. Thus, as a manuscript puts sleep and anesthesia together, there should be more discussion about their similarity and differences.

This is a very important suggestion. Following this suggestion, we now added this discussion (lines 284-289 and 293-299).

2. There is a possibility that terminal stimulation in VTA produces antidromic spikes in BNST cell bodies. The author should exclude such a possibility.

Thanks a lot for this suggestion. We have now added a control experiment, demonstrating that light stimulation of 5 mW or less did not induce antidromic spikes (Supplementary Fig. 5). We used light intensity of 5 mW for axonal stimulation in the experiments.

Reviewer #2 (Remarks to the Author):

Li and colleagues investigated the role of GABAergic neurons in the bed nucleus of the stria terminalis (BNST) in sleep and general anesthesia. They utilized fiber photometry and optogenetic approaches to specifically manipulate BNST GABAergic neurons in Vgat-cre mice to test the sleep-wake transition and EEG alterations under isoflurane general anesthesia.

The authors observed that activation of the BNST GABAergic neurons significantly promoted wake initiation and maintenance, as well reanimating from anesthesia. Moreover, optogenetic results verified that BNST GABA- ventral tegmental area (VTA) neural circuit mediated the wake promotion effect, rather than recovery from general anesthesia.

The experiments are basically well designed and carried out. The conclusion was supported by the data. However, I have some concerns.

We would like to express our gratitude to the reviewer for recognizing the value of our work and the thoughtful feedback and suggestions that are indispensable for the enhancement of our manuscript.

Major comments:

1. Although previous study has revealed that the BNST GABAergic neurons could trigger immediate transition from NREM sleep to wakefulness (Shota et al., J Neurosci 2017), the wake promoting effect of BNST GABAergic neurons in this article seems to be too significant, as both acute and prolonged optical activation could induce over 90% wakefulness. To further confirm the results, you'd better apply optical inhibition and chemogenetics, and carefully discuss about the underlying meaning of the intense wake promoting effect, which may play an important role in stress or psychiatry disorders associated insomnia.

Following your suggestion and that of another reviewer, we conducted a silencing experiment using cell-type specific taCaspase3 to ablate the BNST GABAergic neurons. We confirmed that ablation of BNST GABAergic neurons decreased the amount of wakefulness and increased the amount of NREM and REM sleep during the dark phase (Fig. 5a-f). Our results showed a more significant wake promoting effect of BNST GABAergic neurons. The potential explanation may lie in the use of distinct transgenic animals. The Vgat-Cre mice are a preferable choice for investigating the function of

GABAergic neurons compared to Gad67-Cre mice. According to previous studies, Gad67-Cre transgenic animals might exhibit non-specific labeling of GABAergic neurons (Mickelsen et al., Nat Neurosci, 2019; Judd et al., Elife, 2021). And we also carefully discussed about the underlying meaning of the intense wake promoting effect in stress and psychiatry disorders (lines 258-272).

2. Different from Shota et al. study, the authors found activation of BNST GABAergic neurons could also induce immediate transition from REM sleep to wakefulness. However, in the EEG spectrum, the REM sleep is not typical enough, which may largely influence the reliability of results. The authors should carefully check associated data and results. Following the reviewer's suggestion, we have changed a more typical example in Fig. 3d. We also carefully reviewed associated data and results to ensure the validity and robustness of our findings.

3. The article investigated the effect of BNST GABAergic neurons on general anesthesia, however, there were no description about the behavioral changes.

Following the reviewer's suggestion, we have added behavioral changes in Supplementary Fig. 4, Supplementary Table 1, 2.

4. From Figure 1e, the quantification of calcium activity during wakefulness seems inconsistent with the data from figure 1f.

Apologies for any confusion. We have now standardized the range of y-axis in Fig. 1f to match that of Fig. 1e.

5. You'd better verify that only GABAergic neurons in the BNST were manipulated in this study by using immunohistochemistry double-label staining, and to ensure if the AAV-GCaMP6f and AAV-ChR2 work by immunohistochemistry staining in vivo or whole-cell patch-clamp recordings in vitro.

Following the reviewer's suggestion, we have conducted *in situ* hybridization experiment to confirm that the labeled cells in the BNST were GABAergic neurons (Supplementary Fig. 1b). We have also conducted a control experiment by recording EGFP-labeled BNST GABAergic neurons to exclude motion artifacts in the Ca²⁺ recording experiments (Supplementary Fig. 1c-f). In addition, we have included a set of cell-attached recording data to verify the efficiency of AAV-ChR2 (Fig. 3c).

Minor comments:

1. Mouse gender used for experiments should be stated in the methods. In addition, 8 week old mice used in the study is still in puberty.

8 to 20-week-old mice of both sexes were utilized for the experiment. Mice aged 8-week or older were used for virus injection. And 3-4 weeks after virus expression, mice (at least 11 weeks old) were used for recording and manipulation experiments. Following this suggestion, now we have explained these details in the Methods part.

2. After surgery, if there was any analgesics for the mice?

Yes, after the surgery, analgesics (meloxicam, 1 mg kg⁻¹, Boehringer Ingelheim, Germany) were administered to the mice to manage pain and ensure their well-being.

3. Please note the superscript for calcium, as on line 348.

Sorry for this mistake, and now we have corrected it.

Reviewer #3 (Remarks to the Author):

The neural circuits through which anesthetics suppress consciousness are not yet fully understood. In this study, Li et al. investigated the role of the bed nucleus of the stria terminalis (BNST) GABAergic neurons in isoflurane-induced anesthesia. Using fiber photometry in combination with EEG/EMG recordings to assess the state of consciousness, they found that BNST Ca activity decreased during anesthesia and increased again after isoflurane termination. They also used optogenetics targeting GABAergic BNST neurons to probe the emergence from anesthesia. One key finding is that BNST GABAergic projections to ventral tegmental area (VTA) neurons, likely GABA neurons, have prominent contributions to the control of anesthesia and wakefulness.

Many brain areas/nuclei are involved in the effects of anesthesia. The hypnotic role of VTA GABAergic neurons in sleep-wake regulation and anesthesia has been described in a number of articles published in *Nat. Neurosci.*, *eLife*, and *J. Neurosci.*, and inhibition or reactivation of BNST neurons upstream of the VTA can apparently facilitate recovery from unconsciousness via VTA neurons, likely among many other brain areas. I find it difficult to see the conceptual advance of this paper and wonder whether papers like this one really bring us much closer to a comprehensive understanding of anesthesia.

In recent years, an increasing number of neural substrates associated with inhaled induction and emergence have been identified (Bao, et al., *Curr Biol*, 2023). Nevertheless, the precise involvement of the BNST in the regulation of sleep-wakefulness and anesthesia remains to be elucidated.

The neural circuits that regulate inhaled GA induction and emergence (Bao, et al., *Curr Biol*, 2023).

For the first time, we recorded the calcium activities of BNST GABAergic neurons during sleep-wakefulness cycles in freely moving mice, and also documented their calcium activity under anesthesia. Secondly, previous studies only reported the wake-promoting effect of BNST GABAergic neurons from natural sleep. Here we found that optogenetic activation of BNST GABAergic neurons induced arousal from isoflurane anesthesia. Finally, we also studied the downstream target of these BNST GABAergic neurons on the wake promoting function from natural sleep and general anesthesia, and found this function was partially mediated by BNST-VTA pathway.

The work does not show the extent to which BNST neurons facilitate recovery of consciousness compared with other brain regions, such as the dopaminergic VTA neurons,

which are downstream of BNST neurons and are also likely to be allosterically inhibited by isoflurane during anesthesia and may promote arousal from anesthesia upon reactivation. Kudo et al. reported that the BNST-VTA projection is mainly GABAergic, and the downstream neurons in VTA are mainly GABAergic. Therefore, the activation of BNST-VTA projection is predicted to disinhibit VTA dopaminergic neurons (Kudo et al., *J Neurosci*, 2012). In future studies, we will compare the extent in facilitating recovery of consciousness of BNST GABAergic neurons with that of VTA dopaminergic neurons. However, in this study, we focus on the functions of the BNST GABAergic neurons and their projections to the VTA in the arousal control from natural sleep and anesthesia.

In addition, I have the following specific comments:

- The photometry data show that BNST neurons are primarily active during REM sleep and that activity actually decreases from REM sleep to wakefulness. However, optogenetic activation of these neurons induces wakefulness from NREM or REM sleep. What happens when these neurons are chemogenetically activated?

Kodani et al. have found that chemogenetic activation of BNST GABAergic neurons induced an increase in wakefulness, and a decrease in both NREM sleep and REM sleep in 2 h after CNO injection (Kodani et al., *J Neurosci*, 2017). Together with our optogenetic activation data, these results suggest that activation of BNST GABAergic neurons promotes wakefulness, not REM sleep. The similar phenomenon also exists in other brain regions. The GABAergic neurons in the ventral pallidum, dorsal raphe nucleus, and lateral hypothalamic are both wakefulness and REM sleep active (Li et al., *Mol Psychiatry*, 2021; Cai et al., *Sleep*, 2022; Venner et al., *Curr Biol*, 2016). Yet, chemogenetic activation of these GABAergic neurons could promote wakefulness while diminishing NREM and REM sleep.

- I see no evidence for the physiological importance of BNST neurons for arousal, anesthesia and recovery from anesthesia by performing experiments on loss of function of BNST neurons, e.g. by ablation or inhibition of BNST neurons.

Following this suggestion, we have now added these loss of function experiments in BNST GABAergic neurons. Ablation of these neurons decreased the amount of wakefulness and increased the amount of NREM sleep and REM sleep during the dark phase (Fig. 5a-f, see the effect in induction or emergence of anesthesia in Fig. 5g-h).

- This work would benefit from evidence that isoflurane acts on BNST neurons. How is isoflurane-induced anesthesia, and possibly BNST calcium activity, affected by loss of GABA_A receptor function in the BNST? Can Isoflurane deactivate BNST GABAergic neurons, e.g. in electrophysiology studies?

Yes, we have now provided new electrophysiological data showing that isoflurane deactivated the firing rate of BNST GABAergic neurons in Supplementary Fig. 2. Multiple cellular and molecular mechanisms have been previously found for isoflurane-induced anesthesia, including reduction in neuronal excitability, impairment of action potential conduction, inhibition of Ca²⁺ influx, and suppression of synaptic vesicle exocytosis. (see the review Platholi et al., *Curr Neuropharmacol*, 2022). In particular, GABA_A receptor has been found to be a potential anesthetic target. Isoflurane can directly activate GABA_A receptors by potentiating GABA-induced Cl⁻ currents (Franks, *Nat Rev Neurosci*, 2008; Hu et al., *Nat Neurosci*, 2023). Previous studies have reported that BNST GABAergic neurons

express GABA_A receptors (Dumont et al., J Neurosci, 2004; Romaguera et al., Cell Reports, 2020). Thus, isoflurane may activate GABA_A receptors in BNST GABAergic neurons, and reduce their firing rates and Ca²⁺ activities. In addition, isoflurane can also reduce the excitability and synaptic neurotransmission of BNST GABAergic neurons by other specific mechanism, such as the inhibition of sodium currents (Zhao et al., Anesthesiology, 2019). In the future studies, we will study the detailed mechanism of isoflurane anesthesia on BNST neurons.

Reviewer Comments:

Reviewer #1 (Remarks to the Author):

The authors have thoroughly addressed my previous concerns. I am confident that this manuscript is ready for acceptance.

Reviewer #2 (Remarks to the Author):

No comments.